# Type III ATP synthase is a symmetry-deviated dimer that induces membrane curvature through tetramerization

Rasmus Kock Flygaard [1,2,3], Alexander Mühleip[1,2,3], Victor Tobiasson[1,2] & Alexey Amunts [1,2✉]

Mitochondrial ATP synthases form functional homodimers to induce cristae curvature that is a universal property of mitochondria. To expand on the understanding of this fundamental phenomenon, we characterized the unique type III mitochondrial ATP synthase in its dimeric and tetrameric form. The cryo-EM structure of a ciliate ATP synthase dimer reveals an unusual U-shaped assembly of 81 proteins, including a substoichiometrically bound ATPTT2, 40 lipids, and co-factors NAD and CoQ. A single copy of subunit ATPTT2 functions as a membrane anchor for the dimeric inhibitor $IF_1$. Type III specific linker proteins stably tie the ATP synthase monomers in parallel to each other. The intricate dimer architecture is scaffolded by an extended subunit-*a* that provides a template for both intra- and inter-dimer interactions. The latter results in the formation of tetramer assemblies, the membrane part of which we determined to 3.1 Å resolution. The structure of the type III ATP synthase tetramer and its associated lipids suggests that it is the intact unit propagating the membrane curvature.

[1] Science for Life Laboratory, Department of Biochemistry and Biophysics, Stockholm University, 17165 Solna, Sweden. [2] Department of Medical Biochemistry and Biophysics, Karolinska Institute, 17177 Stockholm, Sweden. [3]These authors contributed equally: Rasmus Kock Flygaard, Alexander Mühleip.
✉email: amunts@scilifelab.se

The F-type ATP synthase ($F_1F_o$-ATP synthase) is responsible for generating most of the cellular adenosine triphosphate (ATP) from adenosine diphosphate (ADP) and inorganic phosphate through rotary catalysis[1,2]. In eukaryotes, ATP synthases are found in the inner mitochondrial membrane. As opposed to the monomeric nature of bacterial[3,4] and chloroplast[5] counterparts, mitochondrial ATP synthases are functional homodimers with a more complex structure, including numerous additional proteins[6–8]. Homodimers of ATP synthase self-assemble into rows, inducing curvature of the inner membrane which is a fundamental prerequisite for cristae formation and essential for mitochondrial biology[9–13]. Electron cryo-tomography (cryo-ET) studies have demonstrated that the architecture of ATP synthase homodimers varies considerably between eukaryotic species[10,14,15]. Due to their key function, these differences in the architecture can also be of clinical relevance[16].

Mitochondrial ATP synthase dimers in different phyla are classified into four types (types I–IV), based on the angle between the peripheral stalks of the monomers and the cristae morphology[16]. Type I dimers are found in animals and fungi, and in situ studies showed that they form the characteristic membrane curvature of lamellar cristae[13]. Type II and IV dimers are present in green alga[17,18] and euglenozoan species[15], respectively, with a ~60° angle between the peripheral stalks. Single particle electron cryo-microscopy (cryo-EM) structures of type I, II, and IV ATP synthase dimers elucidated their subunit composition and organization, as well as provided insight into how the dimers determine the membrane curvature[19–24].

The type III ATP synthase was detected in ciliates where it induces tubulation of the cristae membranes[25]. Cryo-ET studies described an extensive helical row assembly of dimers, suggesting that interdimer interactions might play a role in shaping the membrane[14]. Proteomics analysis detected some specific subunits with no sequence similarity to the other ATP synthase types[26]. However, neither location nor structural function of these subunits are known, and the mismatch with the other types of ATP synthase in forming extensive and stable rows has not been investigated.

In this study, we present the cryo-EM structures of the type III ATP synthase dimer and tetramer with associated lipids from the model organism *Tetrahymena thermophila* in complex with the natural inhibitor $IF_1$. Our structure of the dimer reveals that, unlike in all the previously investigated complexes, the monomers of the membrane-embedded $F_o$-subcomplex are not identical to each other. The commonly observed $C_2$-symmetry is broken by the accommodation of a single ATPTT2 subunit at the dimer interface that anchors the $IF_1$. Subunits that tie the monomers together are distributed between the matrix, transmembrane, and luminal regions, supporting the stable parallel U-shape arrangement. The generation of the membrane curvature is achieved through the tetramerization of the ATP synthase that is supported by subunit-*a*. Together, this work provides the structural basis for the type III ATP synthase parallel arrangement and defines the tetramer as the intact structural unit propagating cristae formation.

## Results

### Structure determination and newly identified elements.

To characterize the type III ATP synthase, we solubilized the mitochondrial membranes with digitonin, and initial analyses using gradient sedimentation and blue-native PAGE confirmed the presence of high-molecular weight species (Supplementary Fig. 1), in line with previous results[26]. The ATP synthase complexes were further purified and subjected to cryo-EM

analysis. 3D classification of ATP synthase particles revealed two populations corresponding to ATP synthase dimer and tetramer. Using masked refinements, the resolution of different regions reached 2.5–3.1 Å (Supplementary Figs. 2 and 3a–e and Supplementary Table 1), allowing de novo modelling. Refinement into a 2.7-Å resolution allowed for construction of a of the ATP synthase dimer. The tetramer was refined to 3.3 Å resolution and the membrane region consisting of interacting $F_o$ subcomplexes was further improved to 3.1 Å resolution upon masked refinement (Supplementary Figs. 2 and 3f). Two copies of the ATP synthase dimer model were refined into the $F_o$-tetramer map generating a composite ATP synthase tetramer model. Both monomers are inhibited in the same rotary state by the natural inhibitor $IF_1$.

The overall structure of the type III ATP synthase dimer has a U-shape (Fig. 1a, b and Supplementary Movie 1), with no other types detected, which is consistent with previous low-resolution studies[14,26]. The ~2 MDa complex comprises 81 protein subunits, 9 of which were not previously reported, and assigned directly from our cryo-EM density map (Supplementary Fig. 4 and Supplementary Table 2). We further identified density in the central stalk corresponding to subunit-ε, which also escaped detection in the previous mass spectrometry analysis[26] (Supplementary Figs. 4 and 9a).

The overall size of the type III ATP synthase dimer is substantially larger than its counterparts. In the $F_o$-subcomplex, 6 out of 7 conserved subunits are extended and 13 subunits have no structural equivalents in other types of ATP synthases (Supplementary Fig. 5). Together, the specific elements add ~0.7 MDa to the type III ATP synthase dimer, doubling its size compared to bacterial counterparts. For the conserved subunits, we adopted the yeast nomenclature[19], and type III specific subunits are named ATPTT1-13 according to descending molecular weight, as previously proposed[21,27] (Supplementary Table 2).

The type III specific subunits are distributed between matrix, membrane, and luminal regions (Fig. 1a, b and Supplementary Fig. 5). In particular, there is an extension in the luminal region, where $F_o$ proteins form a 64 by 160 Å $F_o$-luminal region (Fig. 1c). Consequently, the interface of the monomers facing each other is altered, and a different type of dimerization mode is formed with intricate interactions between monomers (Supplementary Fig. 6). As illustrated by the laterally offset peripheral stalks and conserved subunits (Supplementary Fig. 7), it is essentially different from the yeast, mammalian, and algal ATP synthases[19,20,28]. The specific subunits at the altered interacting surface stably tie the two monomers together in parallel to each other, with little induction of membrane curvature. This differs from the predominant wide-angle V-shape organization of the previously reported mitochondrial ATP synthases, which induce curvature of the nearby membrane (Supplementary Fig. 7).

Apart from the protein subunits, we also assigned in the $F_o$-subcomplex 30 cardiolipins, 6 phosphatidylcholines, 4 phosphatidylethanolamines, and 2 coenzyme Q (CoQ) molecules (Supplementary Fig. 8 and Supplementary Movie 1). Six of the cardiolipins are deeply embedded within the $F_o$-subcomplex, sequestered from the bulk membrane and instead acting as structural components of the ATP synthase dimer structure (Fig. 1e). They are bound in sites equivalent to the matrix and luminal leaflets of the cristae membrane and are likely incorporated during assembly of the $F_o$-subcomplex. Another site of high concentration of native lipids was found around a channel-like cavity extending through the periphery of the $F_o$-subcomplex towards the c-ring (Fig. 1f). This $F_o$ cavity likely provides access for bulk membrane lipids to the c-ring, establishing the luminally sealed matrix proton half-channel required for translocation of protons.

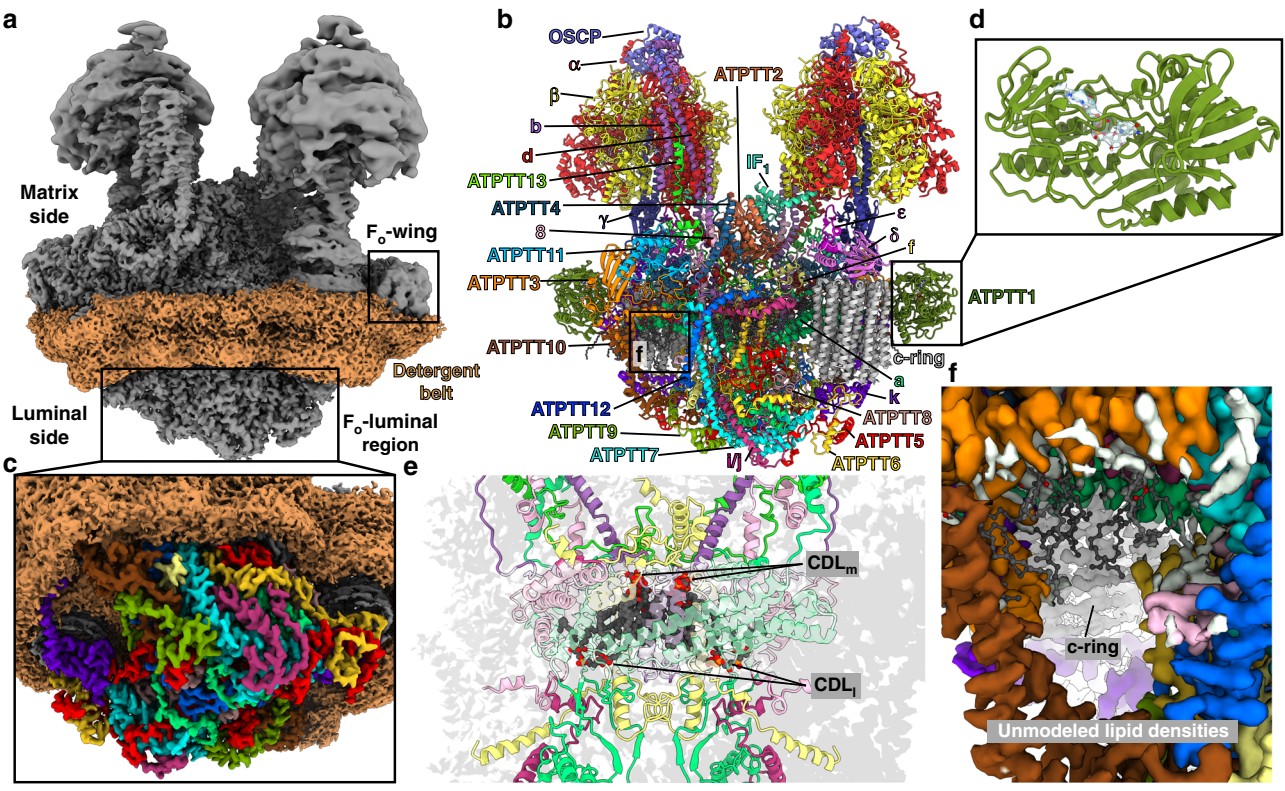

**Fig. 1 Structure of type III ATP synthase dimer with bound native lipids. a** Local-resolution filtered cryo-EM density map of the ATP synthase dimer with detergent belt. **b** Composite model of ATP synthase dimer with colored subunit labels. Black windows highlight regions for close-up views in panels d and f. **c** Cryo-EM density map of type III specific $F_o$-luminal region colored as in **b**. **d** Close-up view of $F_o$-wing subunit ATPTT1 showing the bound NAD co-factor. **e** Cardiolipins sequestered in the $F_o$-subcomplex. $CDL_m$ corresponds to matrix leaflet lipids, $CDL_l$ corresponds to luminal leaflet lipids. **f** $F_o$ cavity extending towards c-ring. Cryo-EM density map is colored by subunits, modeled lipids are shown as stick-balls. Unmodeled lipid-density map features are colored white.

**Characteristics of newly identified subunits**. Among the 13 type III specific subunits, three luminal ATPTT5, ATPTT8, and ATPTT9, adopt coiled-coil-helix-coiled-coil-helix domains (CHCHD). ATPTT5 and ATPTT8 harbor $CX_9C/CX_{10}C$, and ATPTT9 contains $CX_3C/CX_5C$ motifs (Supplementary Fig. 9b). On the matrix side, several conspicuous subunits adopt globular folds, notably ATPTT1-3, and ATPTT11 (Supplementary Fig. 9c).

The $F_o$-wing clamps around the c-ring, and is formed by a 55-kDa subunit ATPTT1 (Fig. 1d). Its tertiary fold superposes well with that of human mitochondrial sulfide:quinone oxidoreductase (Supplementary Fig. 9d), which uses flavin adenine dinucleotide (FAD) for its activity[29]. However, in ATPTT1 the binding pocket formed by two Rossmann folds contains the co-factor nicotinamide adenine dinucleotide (NAD). The nicotinamide group of the NAD is in close proximity to Cys205 (Supplementary Fig. 9e), which could potentially be the active site of ATPTT1. The human sulfide:quinone oxidoreductase is an integral membrane protein, and ATPTT1 locates closely to the matrix leaflet of the crista membrane with several adjacent lipid-like density features resolved in the cryo-EM density map (Supplementary Fig. 9f). Thus, ATPTT1 is positioned to encounter membrane-embedded electron-acceptors, such as CoQ, in a similar fashion as the human enzyme[29].

Further analysis of individual $F_o$ subunits revealed that the C-terminal domain of ATPTT3 adopts a globular fold similar to the intermembrane space domain of the mitochondrial inner membrane import translocase subunit TIM21 (Fig. 1b and Supplementary Fig. 9h). The associated protein subunit ATPTT11

adopts the fold of an acyl-carrier protein, however, no phospho-pantetheine was found conjugated (Fig. 1b and Supplementary Fig. 9i). Unexpectedly, we discovered a Mg-ATP molecule bound as a structural $F_o$-component on the matrix side, serving as a hub of interactions connecting subunits $a$, $k$, ATPTT3 and ATPTT4 (Supplementary Fig. 9k). To our knowledge, this is the first observation of a nucleotide bridging $F_o$-subunits.

The addition of the type III specific protein subunits affects architecture and topology of the conserved subunits. For example, the transmembrane helix of subunit-$i/j$ is tilted to a near-horizontal position as a result of ATPTT7 interaction, while maintaining an interface with subunit-8. Subunit-$b$ contributes its N-terminal helix to the peripheral stalk structure, in addition to the canonical C-terminal helix (Supplementary Fig. 12a). The subunit-$k$ C-terminal extension folds into a four-helical bundle right below the c-ring sealing off its interior (Figs. 1c and 4a).

**Type III ATP synthase forms symmetry-deviated dimer with anchored $IF_1$**. During image processing, we noticed that the quality of the apparent two-fold symmetric density in the central region between the monomers was not consistent with the overall map quality. This region indicated a bilobed shape extending from a lipid-filled central cavity on the matrix side of $F_o$-subcomplex. To improve the map, the aligned particles were subjected to 3D classification, which resulted in their distribution into two major classes with a similar number of particles, related by a 180°-rotation (Supplementary Fig. 2c and "Methods").

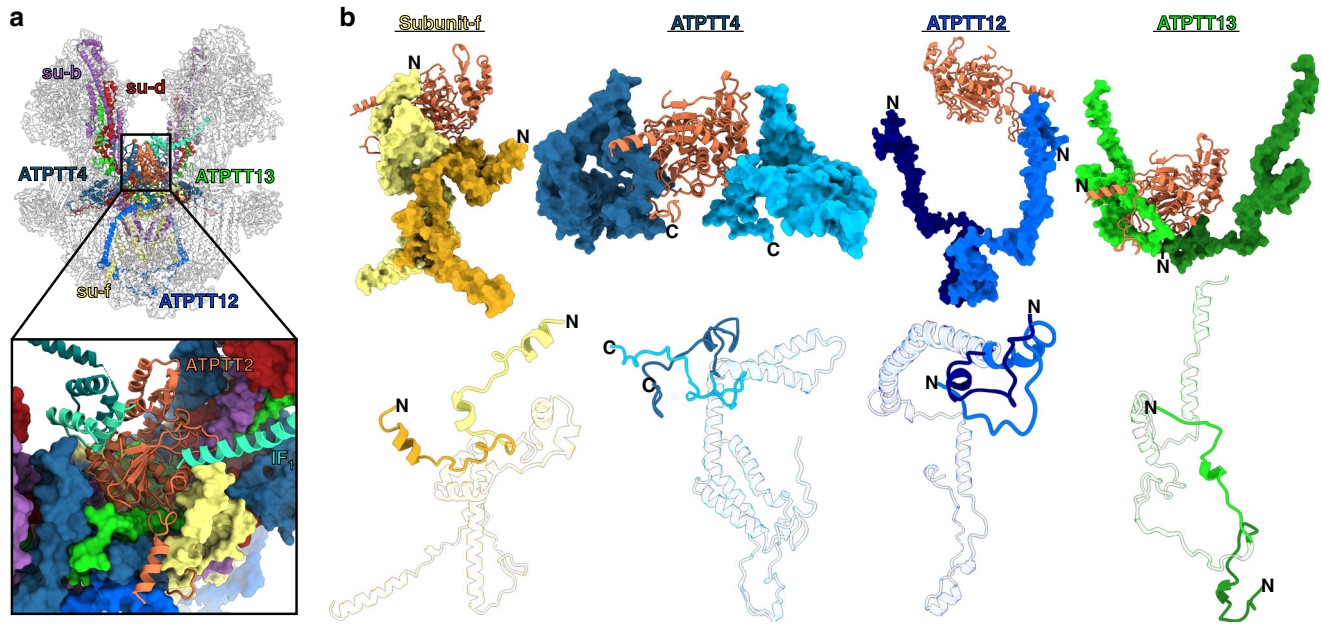

**Fig. 2 The binding of ATPTT2 results in a symmetry-deviated ATP synthase. a** Overview showing the location of the substoichiometric ATPTT2 with its interacting subunits. Close-up view shows the cradle-shaped ATPTT2 binding site (colored surfaces). **b** Symmetry-deviated subunits in respect to ATPTT2. Lower panel: superpositions of subunit from the two monomers features the differences.

Further processing allowed the identification of the involved proteins from the sidechain densities and resulted in a model of three protein subunits clustered in this region. Markedly standing out is ATPTT2, observed only as a single copy in the dimer. This represents the only sub-stoichiometric $F_o$ subunit reported in any ATP synthase dimer. ATPTT2 contains a sequence-conserved core, adopting a classical α/β hydrolase fold, although with an inactive catalytic triad (Supplementary Fig. 9g, j). Homologs of ATPTT2 are conserved in ciliates, indicating that ATPTT2 is a genuine feature of the type III ATP synthase (Supplementary Fig. 10a). The binding site for ATPTT2 is cradle-shaped with subunits $f$ and ATPTT13 lining the membrane-facing interface and subunits $b$, $d$ and ATPTT4 constituting the sides (Fig. 2a).

The association of the ATPTT2 subunit enforces local deviations from the symmetry of the dimer, while the main part of the $F_o$-subcomplex obeys the symmetrical architecture. Comparing the protein subunits interacting with ATPTT2 in the $F_o$-subcomplex features conformational differences between the counterparts (Fig. 2b and Supplementary Movie 1). Subunit-$f$ (residues 2–27) display symmetry mismatch by adopting different folds in the two copies. ATPTT4 C-terminal residues 235–268 differ structurally with one copy wrapping around an extended C-terminal loop of ATPTT2, while the second copy projects away. ATPTT12 (5–29) extend in opposite directions between the two copies, as one copy interacts with the ATPTT2 C-terminal loop enforcing a luminal direction, whereas the second is pointed towards the matrix side. ATPTT13 (21–41) copies interact differently with ATPTT2. Thus, the substoichiometric binding of ATPTT2 induces different conformations in the counterparts from the two monomers resulting in overall symmetry deviation of the type III ATP synthase dimer.

In addition, ATPTT2 has evolved a unique function as a platform for the binding of the natural inhibitor $IF_1$ in a way that inhibits both monomers in the dimer, which was not observed before. Universally, $IF_1$ blocks the rotational movement through its N-terminal domain, while the C-terminal domain is responsible for its dimerization[30,31]. In our structure, the canonical inhibitory helix of $IF_1$ is bound to the $β_{DP}$-subunit of the $F_1$-subcomplex (Supplementary Fig. 11a, b). It is connected to the C-terminus dimerization domain through a flexible (unmodeled) linker (Supplementary Fig. 11c). At the C-terminus, two helices of each $IF_1$ copy form a four-helix bundle, mediated by ATPTT2 (Fig. 3a). The contacts involve hydrophobic and ionic interactions (Fig. 3b, c and Supplementary Fig. 10b). Overall, the type III $IF_1$ dimer is different from the mammalian counterpart that inhibits monomers of neighboring dimers and not of the same dimer (Fig. 3d, e).

**The role of subunit-$a$ and type III specific subunits in ATP synthase dimerization.** The ATP synthase dimer configuration was reported to rely on specific subunits forming a hydrophobic wedge that imposes a local curvature on the membrane[16,32–34]. As previous work showed subunit-$a$ to form dimer contacts[19], we focused our analysis accordingly. Subunit-$a$ is a transmembrane protein critical for proton translocation through the $F_o$-subcomplex[35]. Consistent with the previous studies[4,5,19,36], in our structure it has a six-helix fold with two horizontal helices $H5_a$ and $H6_a$ contacting the c-ring, while the remainder part of the protein interacts with the conserved subunits $b$, $d$, $f$, $i/j$, $k$, 8, and a non-conserved interaction with subunit-$b'$ (Fig. 4a and Supplementary Fig. 12b). However, the dimer contacts are established heterotypically between residues 362-364 in the loop connecting helices $H5_a$ and $H6_a$ in one subunit-$a$ copy and the N-terminal extension on the other (Fig. 4b).

In addition, the type III ATP synthase subunit-$a$ has 170- and 38-amino acid N- and C-terminal extensions, respectively. The overall length of 446 amino acids makes it by far the largest subunit-$a$ homolog described in any F-type ATP synthase (Fig. 4c). The N-terminal extension first folds into a four-helix bundle, which interacts with subunits $i/j$ and ATPTT5-8 on the periphery of the luminal $F_o$-region (Fig. 4d) and further threads across the symmetry axis of the $F_o$-subcomplex interacting with symmetry-related subunits $a'$, ATPTT5', 7'–9', 12' (Supplementary Fig. 12c).

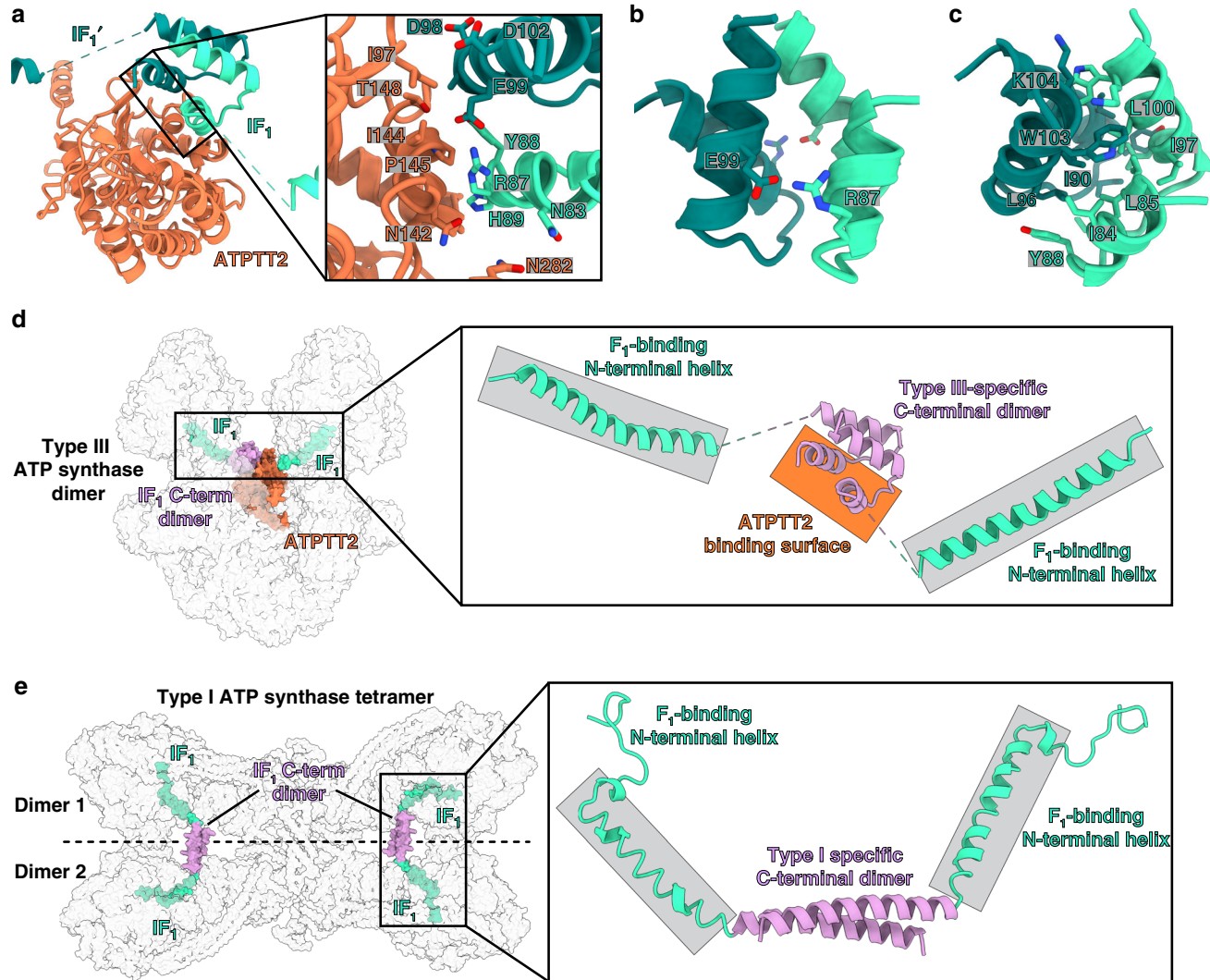

**Fig. 3 ATPTT2 anchors IF₁ to the membrane. a** ATPTT2 interacts with the IF₁ C-terminal domains, which dimerizes via their helical hairpins. Close-up view shows the interaction interface between ATPTT2 and IF₁ C-terminal dimer. **b** IF₁ C-terminal interact via electrostatic interactions on the periphery, and **c** form a tight hydrophobic core. **d** Type III ATP synthase dimer with bound intra-dimeric IF₁. The elements forming C-terminal dimer, interact with ATPTT2 and F₁ are indicated in right panel. **e** porcine type I ATP synthase tetramer (PDB 6J5K) [https://www.rcsb.org/structure/6J5K] with bound inter-dimeric IF₁.

Thus, subunit-*a* ties the entire F₀-subcomplex together in a rigid structure and serves a central function of scaffolding the surrounding subunits to position them for interactions.

Additional type III specific dimerization motifs were identified in our structure throughout the entire span of the interface (Supplementary Fig. 6). The total surface area buried between the monomers exceeds 16,000 Å², which is at least four times more than for the characterized other types. We found contacts in the membrane (subunits *a*, *b*, and *f*), matrix (subunit-*f* and ATPTT13) and lumen (subunits *a*, *f*, *k*, ATPTT5-9, ATPTT12) (Fig. 5a and Supplementary Fig. 6 and 12d–j). In addition to homotypic contacts along the symmetry axis, it is further supported by a series of extensive heterotypic contacts located up to ~60 Å off the dimerization axis (Fig. 5a, b). The heterotypic contacts involve a cardiolipin molecule bound inside the c-ring bridging subunit-*k* and ATPTT6′. Homotypic subunit-*b* contacts also involve bound cardiolipins (Fig. 5c), whereas subunit-*f* interface involves protein contacts only (Fig. 5d). Consequently, the type III ATP synthase dimer structure contains by far the most extensive dimer interface observed in any reported ATP synthase dimer.

**Type III ATP synthase assembles into tetramers via subunit-*a* and luminal contact sites.** Since the type III ATP synthase forms a U-shaped dimer that does not induce membrane curvature, we further investigated how dimers assemble into higher oligomers by focusing on tetrameric particles. We used masked refinement to obtain a 3.1-Å resolution map of the tetrameric membrane region, comprising two dimeric units (Fig. 6a and Supplementary Figs. 2c and 3f). To interpret the interface between two dimers, we fitted the dimer models and refined them into the F₀ tetramer density map (Fig. 6b and Supplementary Fig. 13a). All protein densities could be explained by subunits of the dimer models, and no additional protein moieties appear to be needed to hold the tetramer together. The tetrameric model fits a previously reported subtomogram average of the ciliate ATP synthase[14] (Supplementary Fig. 13b), suggesting that the tetramer is representative of the native configuration of the type III ATP synthase.

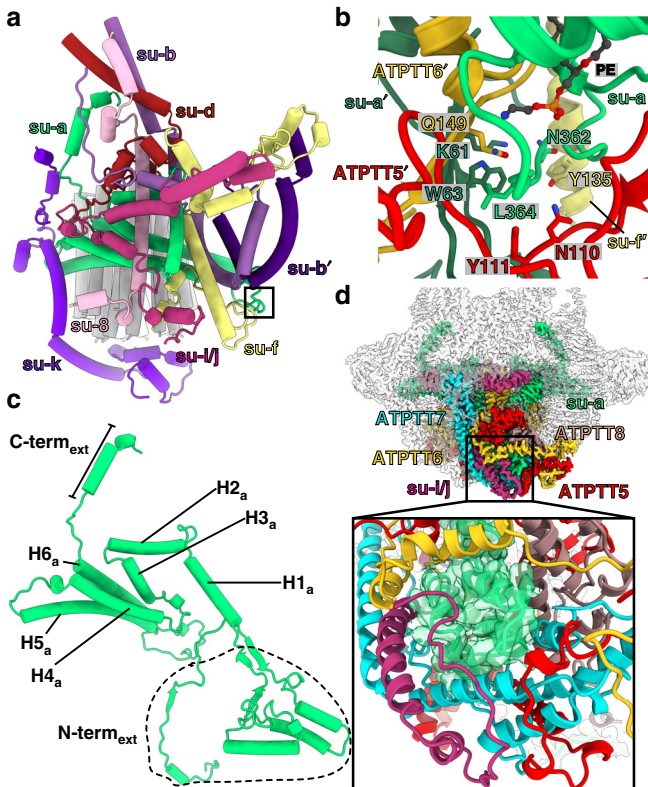

**Fig. 4 Subunit-*a* interactions throughout the F$_o$-subcomplex. a** Subunit-*a* canonical six-helical core surrounded by the conserved subunits *b, d, f, i/j, k* and 8. It forms a non-canonical interaction with subunit-*b'*. The c-ring is grey. Window corresponds to the region shown in **b** close-up view.
**b** Subunit-*a* forms a dimer interface with subunits *a', f'*, ATPTT5-6', with a phosphatidylethanolamine bound close to the interaction site. **c** Close-up of subunit-*a*, showing the six-helical core fold (H1-H6$_a$) with N- and C-terminal extensions a labeled. **d** Overview of F$_o$-subcomplex cryo-EM density showing peripheral location of subunit-*a* N-terminal extension and interacting subunits (subunit-colored, rest transparent gray). Close-up view shows subunit-*a* N-terminal extension scaffolding of helices from other F$_o$-subunits.

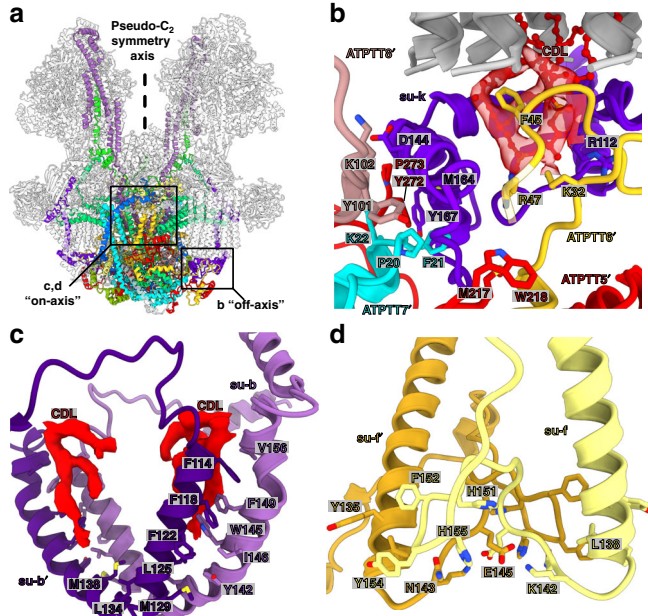

**Fig. 5 Dimer interfaces are scattered throughout the F$_o$-subcomplex. a** Overview of ATP synthase dimer structure with F$_o$ subunits participating in dimer interfaces shown with their respective colors. Windows indicate regions shown in panels b-d close-up views. **b** The extended C-terminal domain of subunit-*k* engages in several dimer interface interactions below the c-ring with F$_o$ subunits ATPTT5-8'. A bound cardiolipin (CDL, red) mediates interactions between subunit-*k* and ATPTT6'. **c** Subunits *b* and *b'* form a hydrophobic dimer interface with cardiolipins (CDL, red). Cardiolipin density is carved 2.6 Å around atoms in F$_o$-dimer local-resolution filtered map. **d** Subunits *f* and *f'* form a dimer interface through electrostatic, polar and hydrophobic interactions.

In the tetramer, the ATP synthase dimers are spaced 140 Å center-to-center with an angle of ~11° between dimer axes bringing the F$_o$-subcomplexes together (Fig. 6c, left panel). In addition, along a perpendicular axis parallel to the membrane plane, the dimers are arranged with a ~3° angle (Fig. 6c, right panel). This angular arrangement induces curvature of the surrounding membrane in a right-handed helical shape, which is also evident from the detergent/membrane density surrounding the ATP synthase tetramer (Supplementary Fig. 13c). Applying multi-body refinement analysis of the ATP synthase tetramer, using the two dimers as bodies, indicated no distinct conformational states but instead revealed continuous motions in a complicated motion landscape (Supplementary Fig. 14a). The analysis showed that the intrinsic motion is dominated by combinations of tilting and rotation motions of one dimer with respect to the other (Supplementary Fig. 14b–d). This agrees with previous observations[14] and reflects the need for plasticity in type III ATP synthase oligomers during membrane tubulation.

The protein-protein interactions holding the two dimers together can be classified into two interface types: direct interactions between F$_o$ subunits in the lumen, and interactions mediated by lipids (Fig. 6b). The luminal interface is governed by homotypic interactions between subunit-*i/j* and ATPTT6 located in neighboring F$_o$-subcomplexes. The extended subunit-*a* N-terminal four-helix bundle positions ATPTT6 with a horizontal α-helix to interact with the ATPTT6 subunit in the other ATP synthase dimer (Fig. 7a). This luminal interface is dominated by electrostatic interactions between arginine and glutamate residues as well as stacking interactions between arginines (Fig. 7b). Further into the luminal side, the subunit-*i/j* copies are positioned by the N-terminal subunit-*a* extension to form an extensive interface where residues Asn207 stack against each other and residues Ser173, Lys174 and Ser198 are within distances enabling polar interactions between main and sidechains (Fig. 7a, c).

The second type of interface involves lipids bound between the two dimers. Most conspicuous is the interface located between copies of subunit-*i/j* and ATPTT5 (Fig. 6b). The gap between the subunits in the F$_o$-subcomplexes is just wide enough to accommodate three lipids between the two ATP synthase dimers (Fig. 7d). The two subunit-*i/j* copies interact with headgroups of membrane lipids in the matrix leaflet through polar and charged residue sidechains extending from two α-helices arranged in-plane with the matrix leaflet headgroups. The narrow, lipid-filled gap in the luminal leaflet is established by the two ATPTT5 copies, which display horizontal α-helices, in-plane with the luminal leaflet headgroups and contain polar residue sidechains enabling protein-lipid interactions (Fig. 7d). Lipid-mediated interfaces are also found more peripherally in the ATP synthase tetramer, where F$_o$ subunits ATPTT3, ATPTT10, and ATPTT12 on one side clamp membrane lipids against the c-ring on the second ATP synthase dimer (Supplementary Fig. 13d). Thus, type

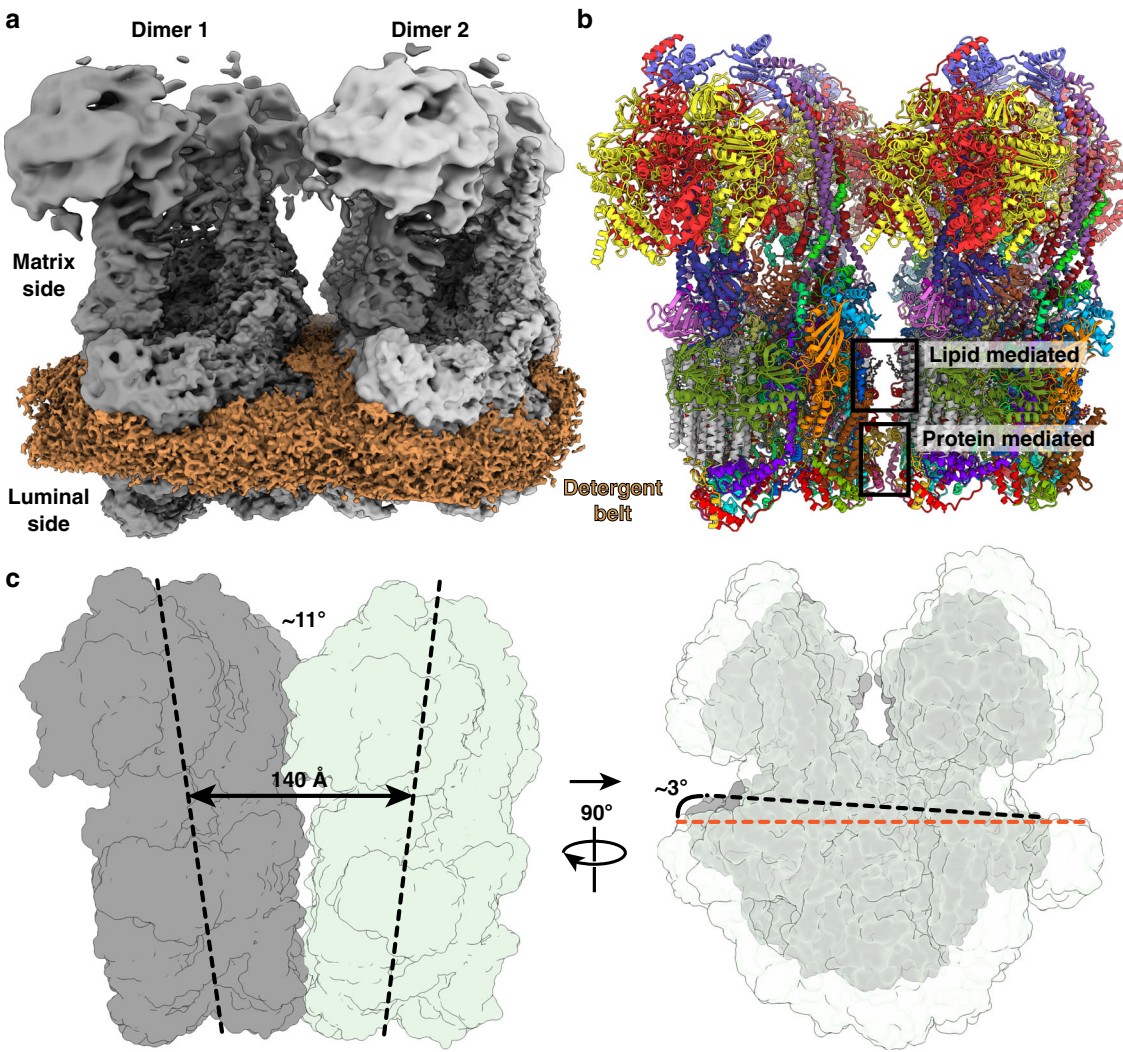

**Fig. 6 Structure of type III ATP synthase tetramer. a** Local-resolution filtered cryo-EM density map of ATP synthase $F_o$ tetramer. Dimer 1 is colored grey and dimer 2 colored light gray. **b** Model of ATP synthase tetramer. Boxes indicate protein and lipid interaction interfaces. **c** ATP synthase dimers spaced by 140 Å within tetramers are angled by ~11° (sideview, left) and ~3° (end-view, right), generating membrane curvature.

III specific subunits together with lipids play a crucial role in the architecture of the ATP synthase tetramer by linking dimers.

## Discussion

Bioenergetics depends on a supporting membrane bilayer, whose shape can be adapted to optimize the organization of the energy conversion process. In mitochondria, this function is performed by $F_1F_o$-ATP synthase dimers. In this study, we characterized the largest ATP known synthase dimer, with multiple components that have no equivalents in other counterparts, including protein subunits, co-factors, and lipids.

The analysis of the proteins revealed 13 type III specific subunits, 4 of which adopt tertiary folds of known enzymes, such as ATPTT1 with a sulfide:quinone oxidoreductase fold and NAD co-factor. Acquisition of abundant catalytic enzymes to perform structural roles have been previously reported for soluble mitochondrial complexes[37,38]. Here we show that this mechanism also takes place in the membrane subunits of the ATP synthase that are formed from precursors of proteins that were pre-existing in the mitochondrial compartment.

The analysis of co-factors revealed two CoQ molecules embedded in the $F_o$-subcomplex. In mitochondria, CoQ is a mobile electron carrier[39] that laterally diffuses in the membrane[40]. It performs the

function of a mediator in the electron transfer reaction and was shown to transiently bind to complexes I, II, and III[41–43]. However, >84% of CoQ is estimated to diffuse freely in the bilayer[44], and non-exchangeable CoQ molecules have not been reported in mitochondria before[45]. We find that type III ATP synthase tightly binds CoQ within the protein scaffold, and its function remains an open question.

The analysis of natively bound lipids showed that cardiolipins are by far the most predominant. A key cardiolipin was observed bound in the dimer interface between subunit-$k$ and ATPTT6′, with the four acyl chains extending to the hydrophobic interior of the c-ring. This observation of lipid inside the c-ring agrees with previous studies observing lipid plugs or detergents, mimicking lipids, inside isolated c-ring complexes[46,47]. Likewise, atomistic molecular dynamics simulations suggested the presence of numerous lipids inside spinach c-rings accounting for unmodeled cryo-EM density features[5,48]. Thus, together with similar results for a V-type ATPase[49], our observation shows a preference for cardiolipin binding inside c-rings. This highlights a subunit-mediated selection mechanism for this lipid, as cardiolipin is not the most prevalent lipid in *T. thermophila* mitochondria[50].

Our structure also revealed a substoichiometric component ATPTT2 that causes a deviation from the two-fold rotational

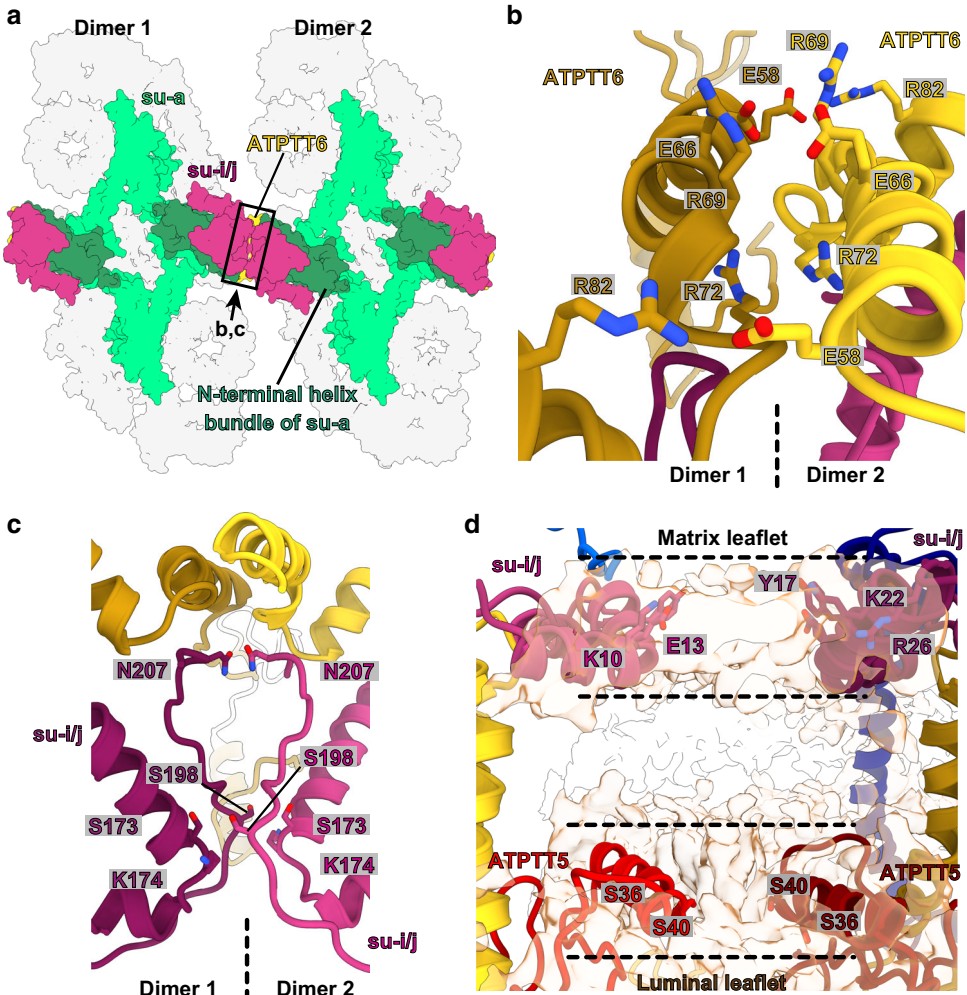

**Fig. 7 Type III ATP synthase tetramer interacts via specific protein and lipid-mediated interfaces. a** Luminal view of the tetramer. The N-terminal four-helix bundle of subunit-*a* (dark green) positions subunit-*i/j* and ATPTT6 for tetramer interactions. Boxed region and arrow indicate region and view direction shown in close-up view in **b**, **c**. **b** ATPTT6 subunits in the two dimers form electrostatic interactions. **c** Subunits-*i/j* in the interface form polar interactions. **d** Subunits *i/j* and ATPTT5 interact via bound membrane lipids in the matrix leaflet (black dashed lines) and luminal leaflet (gray dashed lines). Unmodeled lipid density is shown in transparent light brown color.

symmetry. ATPTT2 is a ~43-kDa protein with an asymmetric α/β-hydrolase fold, that provides a binding site for the C-terminal helix bundle of the IF$_1$ dimer. IF$_1$ dimers have been reported to be the inhibitory form in mammals and *Trypanosoma*[30,51,52]. The mammalian IF$_1$ dimer forms via a long antiparallel α-helical coiled coil in the C-terminal region. By forming dimers of dimers, the inhibitor is thought to remain inactive as an IF$_1$ tetramer until pH-dependent dissociation into dimers triggers inhibition[30,51]. Due to its markedly different C-terminal four-helix bundle, the ciliate IF$_1$ dimer does not form the mammalian-type IF$_1$ tetramer. Instead of depending on a pH switch, dimeric IF$_1$ may be bound permanently or be specifically recruited to F$_o$ via ATPTT2, providing spatial proximity to F$_1$ to allow inhibition of ATP hydrolysis. Given that the primary structures of ATPTT2 and IF$_1$ are conserved in ciliate organisms, the inhibitory mechanism is likely a conserved feature in this phylum. Thus, the type III structure indicates that not only ATP synthase, but also IF$_1$ dimerization may have evolved independently between mitochondrial lineages.

The type III ATP synthase dimer is arranged with the two parallel monomers, imposing little induction of membrane curvature. Central to the dimerization is subunit-*a*, where the extensions contribute heterotypic contacts and a binding surface for other protein subunits from the associated monomer. Thus, the dimerization mechanism in type III ATP synthase is different from the type I, II, and IV dimers and depends on newly identified structural elements that are specific to this system (Supplementary Figs. 5–7).

The membrane curvature is induced by the type III ATP synthase assembling into tetramers. The analysis of the interface forming the tetramer revealed that the specific N-terminal helical bundle of subunit-*a* functions as an adaptor for subunit-*i/j* and ATPTT6 that are positioned to engage in interactions associating two neighboring dimers in a tetramer. By extrapolation, the same interface is available for the addition of multiple dimer modules, all with an angular arrangement represented by the type III ATP synthase tetramer structure. This produces a right-handed helical row with 36 ATP synthase dimers per helical turn, which underpins the organization of the inner mitochondrial membrane into tubular cristae (Fig. 8). These data explain the formation of tubulated cristae in ciliates[14].

Together, our study provides molecular details of the most unusual modulation of the ATP synthase that expands the understanding of the relationship between this key bioenergetic system and crista shape.

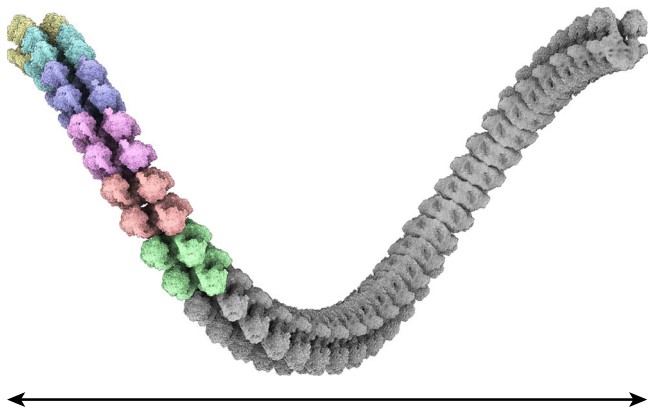

**36 F$_1$F$_o$-ATP synthase dimers**

**Fig. 8 Helical row model of type III ATP synthase extrapolated from the single-particle reconstruction.** Six copies of ATP synthase tetramer (colored densities) arranged by superposition according to the inherent architecture (using a bridging tetramer map, not shown) results in a row with right-handed helical twist. Two-fold propagation of the six ATP synthase tetramer copies (gray colored copies) completes a helical row with 36 ATP synthase dimers in one helical turn.

## Methods

**Isolation of *T. thermophila* mitochondria.** Cells were cultivated in medium consisting of 10 g/L peptone and 5 g/L yeast extract supplemented with 3% v/v glycerol and 10 μM FeCl$_3$. Cultivation temperature was 36 °C with orbital shaking for aeration of cultures. All subsequent steps were carried out at 4 °C or on ice. Late log-phase cells with a density of ~0.8 × 10$^6$ cells/mL were harvested by centrifugation at 1300 × $g$ for 10 min. Cell pellets were resuspended in homogenization buffer (20 mM Hepes/KOH pH 7.5, 350 mM D-mannitol, 5 mM EDTA, 1x protease-inhibitor tablet) and lysed in a Dounce homogenizer kept on ice. Lysate was cleared by centrifugation at 1300 × $g$ for 15 min. The supernatant was filtered through cheesecloth and raw mitochondria were pelleted by centrifugation at 7000 × $g$ for 20 min. Crude mitochondrial pellets were resuspended in buffer SEM (20 mM Hepes/KOH pH 7.5, 250 mM sucrose, 1 mM EDTA). Mitochondria were isolated on discontinuous sucrose gradients prepared with 15%, 23%, 32%, and 60% w/v sucrose in buffer SEM. Gradients were prepared in SW28 tubes and mitochondria sedimented by ultracentrifugation at 141,371 × $g$ for 60 min. Mitochondria sedimented in the interface between 32% and 60% sucrose solutions were aspirated into Falcon tubes, snap-frozen in liquid nitrogen and kept at −80 °C for storage.

**Purification of *T. thermophila* ATP synthase.** All steps were carried out at 4 °C or on ice. Frozen mitochondria were thawed on ice and solubilized in lysis buffer (25 mM Hepes/KOH pH 7.5, 25 mM KCl, 5 mM MgCl$_2$, 4% w/v digitonin) for 1 h on ice. Lysate was cleared by centrifugation at 30000 × $g$ for 30 min. For initial analysis of solubilization efficiency, cleared lysate was layered on top of a sucrose gradient (25 mM Hepes/KOH pH 7.5, 25 mM KCl, 5 mM MgCl$_2$, 0.1% w/v digitonin) prepared with 15–50% w/v sucrose in SW40 tubes. Gradients were centrifuged for 20 h at 90,000 × $g$ and fractionated top-to-bottom into 500 μL fractions. The sucrose gradient fractions were analyzed on blue-native PAGE (see below). With verification of the efficiency of the mild solubilization procedure, the purification protocol started with placing cleared lysate on a sucrose cushion (25 mM Hepes/KOH pH 7.5, 25 mM KCl, 5 mM MgCl$_2$, 0.1% w/v digitonin, 30% w/v sucrose) prepared in Ti70 tubes and centrifuged at 164,685 × $g$ for 3 h. The resulting pellet was gently washed in buffer D (25 mM Hepes/KOH pH 7.5, 25 mM KCl, 5 mM MgCl$_2$, 0.1% w/v digitonin) before resuspended fully in buffer D. The resuspended pellet material was cleared of aggregates by centrifugation at 30,000 × $g$ for 20 min. Cleared supernatant was loaded onto a Superose 6 Increase 3.2/300 column pre-equilibrated in buffer D. Elution fractions of 100 μL were collected throughout the run. Freshly gel filtrated protein sample was used for preparation of grids.

**Blue-native PAGE analysis.** Sucrose gradient fractions were analyzed for high-molecular weight species using 3–12% Bis-Tris native-PAGE gels (Invitrogen). Fraction samples were mixed with 4:1 v/v with 5% G-250 sample additive (Invitrogen). Gels were run at 150 V constant for 90 min at 4 °C using dark-blue cathode buffer additive (Invitrogen). Gels were destained in 8% v/v acetic acid.

**Preparation of cryo-EM grids and data collection.** ATP synthase elution fractions from the gel filtration run were of a concentration of ~10 mg/mL and aliquots were diluted in buffer D to 0.75 mg/mL immediately before application onto grids. A home-made amorphous carbon layer ~3 nm thick was floated onto Quantifoil

R2/2 300-mesh grids and grids were glow-discharged immediately before sample application. A sample volume of 3 μL was applied onto grids and vitrified by plunge-freezing into liquid ethane cooled by liquid nitrogen using a Vitrobot Mark IV. A wait time of 30 s was used with a blot time of 3 s. Micrograph data was collected using EPU 1.9 on a Titan Krios (ThermoFisher Scientific) operated at 300 kV at a nominal magnification of 165 kx (0.83 Å/pixel) with a Quantum K2 camera (Gatan) using a slit width of 20 eV. With an objective lens aperture of 70 μm, images were collected with an exposure rate of 4.26 electrons/pixel/second with 5 s exposure fractionated into 20 frames. A total of 15439 movies were collected.

**Image processing.** Movies were motion-corrected using the RELION implementation of the MotionCor2 algorithm[53,54]. Contrast-transfer function parameters were estimated using CTFFIND4[55]. All subsequent image processing steps were done in RELION-3.1[56] (Supplementary Fig. 2). Particles were picked using reference-based auto-picking and the particle set was cleaned by 2D classification and further by 3D classification. ATP synthase tetramer particles from 3D classification were pooled and refined without symmetry against a reference with correct handedness. 2D classification without image alignment was used to discard the remaining junk or bad particles resulting in 40,691 good particles extracted unbinned in a 600-pixel box. Application of C$_2$ symmetry together with iterative refinements, CTF refinement with higher-order aberration corrections and Bayesian polishing resulted in a reconstruction of the ATP synthase tetramer at 3.3-Å resolution. Masked refinement of the tetramer F$_o$-subcomplexes resulted in a reconstruction resolved to 3.1 Å with local resolutions extending to 2.7 Å (Supplementary Fig. 3). From the initial 3D classification, two classes showed distinct ATP synthase dimer structures. A total of 67,540 particles from these two classes were pooled, extracted, unbinned in a 600-pixel box and refined to a resolution of 2.8 Å after successive rounds of CTF refinement and Bayesian polishing. To resolve the matrix-exposed F$_o$ middle region, 3D classification ($T = 10$) using local angular searches on subtracted particles were performed and resulted in two major classes related by a 180°-rotation of the central density. To overcome this, the initial ATP synthase dimer particles were refined against a reference, aligned to C$_2$ symmetry axis, but without application of symmetry during refinement. The newly refined structure was again used to local-angle 3D classify ($T = 10$) signal-subtracted particles on the F$_o$ middle region. Non-subtracted particles from one of the two major classes were then refined against the C$_2$ symmetry axis aligned reference and the new, refined particles were symmetry expanded, taking only the rotated particles into further processing. The original particles from one 3D class were then joined with the rotated particles of the second 3D class resulting in a total of 61,157 particles. This new set of ATP synthase dimer particles was C$_1$-refined together against the C$_2$-aligned reference, thus still not applying symmetry. The resulting reconstruction went to 2.7-Å resolution now with the F$_o$ middle region well resolved. From this 3D consensus refinement of the ATP synthase dimer, masked refinements were performed of the F$_o$-subcomplex, the F$_o$-wing region, the F$_1$/peripheral stalk, and the central stalk/c-ring (Supplementary Fig. 2c). In none of the masked refinements was the set of dimer particles symmetry expanded. The masked refinements all improved the resolutions of the respective regions, with the F$_o$-subcomplex resolved to 2.5 Å, the F$_o$-wing region resolved to 2.8 Å, the F$_1$/peripheral stalk resolved to 2.9 Å and the central stalk/c-ring resolved to 3.1 Å (Supplementary Fig. 1d). Local resolutions for all reconstructions were calculated using RELION-3.1 and can be found in Supplementary Fig. 3. For multi-body refinement of the ATP synthase tetramer, particles were down-scaled to a pixel size of 1.9453 and extracted into a 256-pixel box. The two dimers were masked as the two bodies using initial angles from the F$_o$-subcomplex masked refinement.

**Model building and refinement.** All manual model building was done in Coot[57]. F$_1$ subunits α, β, γ, δ and oligomycin sensitivity conferring protein (OSCP) N-terminal domain together with F$_o$ subunit c were homology modeled[58] using a yeast reference model (PDB 6CP6)[59] [https://www.rcsb.org/structure/6CP6] and fitted to *T. thermophila* ATP synthase reconstructions from masked refinements of the F$_1$/peripheral stalk and c-ring/central stalk. The mask refined c-ring/central stalk map showed unambiguous density for the presence of F$_1$ subunit ε, which was modeled de novo and verified by BLAST search against a mass-spectrometry-generated list of proteins present in our purified sample. A similar approach was taken for identification and building of OSCP C-terminal domain and F$_1$ inhibitor protein IF$_1$ N-terminal helix, which were both unambiguously resolved in the F1/peripheral stalk density map and detected in the mass spectrometry data. All F$_o$ subunits, including the IF$_1$ C-terminal dimerization domain, were modeled de novo in the high-resolution density map from the F$_o$-mask refined reconstruction, or the F$_o$-wing mask refined reconstruction. Protein subunits were identified by building stretches of peptides and BLAST searching the built sequences against the mass spectrometry data or cross-validated to a published list of putative subunits[26]. This approach allowed us to identify and build 20 unique F$_o$ subunits of which 8 had not previously been associated as ATP synthase subunits (Supplementary Table 2). Resolved lipids, phosphate ions, metal ions and ligands such as ATP, ADP, and NAD were manually fitted into the density map features and restraints for refinement generated using elbow in PHENIX[60]. The resolved coenzyme Q (CoQ) molecules were modeled with 8 isoprenyl units based on previously published analyses made in *T. thermophila*[61]. ATPTT1, including its NAD ligand, was refined separately in the high-resolution F$_o$-wing map using PHENIX real-space refinement[62]. The refined ATPTT1 model was merged with the remainder of

the $F_o$ model and real-space refined using secondary structure restraints. $F_1$/peripheral stalk and c-ring/central stalk models were likewise real-space refined against their respective high-resolution maps using secondary structure restraints. All individually refined models were combined into a composite ATP synthase dimer model and real-space refined against the full ATP synthase dimer reconstruction using reference restraints. For atomic model generation of the ATP synthase tetramer, two dimer models were refined into the $F_o$-tetramer map using reference restraints. Of the three possible tetramer conformations, we modeled conformation 1 where the dimers are related by a translation operation. The tetramer interface is not affected by the pseudo-symmetric nature of the dimers, thus the analysis of interactions is valid for all three tetramer conformations. Model statistics were generated using MolProbity[63] and EMRinger[64] (Supplementary Table 1). Overfitting of atomic models was estimated by off-setting atomic coordinates using "Shake" in CCP-EM[65], real-space refining those models into one of the two unfiltered half maps, respectively, and calculating $FSC_{work}$ and $FSC_{test}$ curves as described[66] (Supplementary Fig. 3). All figures were prepared using Chimera[67], ChimeraX[68], and PyMOL 2.0 (Schrödinger). The DALI[69], PISA[70], and ConSurf[71] servers were used for structural analysis of tertiary folds, buried surfaces, and conservation, respectively.

**Mass spectrometry analysis**. The gel filtrated ATP synthase sample was solubilized in 1% w/v SDS lysis buffer and prepared for mass spectrometry analysis using a modified version of the SP3 protein clean up and digestion protocol[72], followed by SCX peptide clean up. In brief, 70 µg protein from each sample was alkylated with 4 mM chloroacetamide. Sera-Mag SP3 bead mix (20 µl) was transferred into the protein sample together with 100% v/v acetonitrile to a final concentration of 70% v/v. The mix was incubated with rotation at room temperature for 18 min. The mix was placed on magnetic rack and supernatant discarded, followed by two washes with 70% v/v ethanol and one with 100% v/v acetonitrile. The beads-protein mixture was reconstituted in 100 µl LysC buffer (0.5 M urea, 50 mM Hepes/KOH pH 7.6 and 1:50 enzyme (LysC) to protein ratio) and incubated overnight. Finally, trypsin was added in 1:50 enzyme to protein ratio in 100 µl of 50 mM Hepes pH 7.6 and incubated overnight. The peptides were eluted from the mixture followed by peptide concentration measurement (Bio-Rad DC Assay). LC-ESI-MS/MS Q-Exactive Online LC-MS was performed using a Dionex UltiMate™ 3000 RSLCnano System coupled to a Q-Exactive mass spectrometer (ThermoFisher Scientific). The peptides were trapped on a C18 guard desalting column (Acclaim PepMap 100) and separated on a long C18 column (Easy spray PepMap RSLC). The nanocapillary solvent A was 95% water, 5% DMSO, 0.1% formic acid; and solvent B was 5% water, 5% DMSO, 95% acetonitrile, 0.1% formic acid. At a constant flow of 0.25 µl min$^{-1}$, the curved gradient went from 6% B up to 43% B in 180 min, followed by a steep increase to 100% B in 5 min. FTMS master scans with 60,000 resolution (and mass range 300–1500 m/z) were followed by data-dependent MS/MS (30,000 resolution) on the top 5 ions using higher energy collision dissociation (HCD) at 30% normalized collision energy. Precursors were isolated with a 2 m/z window. Automatic gain control (AGC) targets were $1 \times 10^6$ for MS1 and $1 \times 10^5$ for MS2. Maximum injection times were 100 ms for MS1 and MS2. The entire duty cycle lasted ~2.5 s. Dynamic exclusion was used with 60 s duration. Precursors with unassigned charge state or charge state 1 were excluded. An underfill ratio of 1% was used. For peptide and protein identification, the MS raw files were searched using Sequest-Percolator under the software platform Proteome Discoverer 1.4 (ThermoFisher Scientific) against *T. thermophila* protein database and filtered to a 1% FDR cut off. We used a precursor ion mass tolerance of 10 ppm, and product ion mass tolerances of 0.02 Da for HCD-FTMS. The algorithm considered tryptic peptides with maximum 2 missed cleavage; carbamidomethylation (C) was set as fixed modification and oxidation (M) as variable modification.

**Reporting summary**. Further information on research design is available in the Nature Research Reporting Summary linked to this article.

## Data availability
Cryo-EM maps have been deposited at the EMDB with the following accession codes: EMD-10857 ($F_o$-wing), EMD-10858 (c-ring/central stalk), EMD-10859 ($F_o$ dimer), EMD-10860 ($F_1F_o$ dimer), EMD-10861 ($F_o$ tetramer), and EMD-10862 ($F_1$/peripheral stalk). The atomic coordinates have been deposited at the PDB with the following accession codes: 6YNV ($F_o$-wing), 6YNW (c-ring/central stalk), 6YNX ($F_o$ dimer), 6YNY ($F_1F_o$ dimer), 6YNZ ($F_o$ tetramer), and 6YO0 ($F_1$/peripheral stalk).

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

## Acknowledgements

This work was supported by the Swedish Foundation for Strategic Research (FFL15:0325), Ragnar Söderberg Foundation (M44/16), Swedish Research Council (NT_2015-04107), Cancerfonden (2017/1041), European Research Council (ERC-2018-StG-805230), Knut and Alice Wallenberg Foundation (2018.0080), EMBO Young Investigator Program. The data was collected at the SciLifeLab cryo-EM facility funded by the Knut and Alice Wallenberg, Family Erling Persson, and Kempe foundations. Mass spectrometry analysis was performed by the SciLifeLab proteogenomics core facility. We thank Linnea Axelsson for assisting.

## Author contributions

R.K.F., A.M., and A.A. designed the project. R.K.F., A.M., and V.T. prepared the sample and collected cryo-EM data. R.K.F. and A.M. processed the data. R.K.F. and A.M. built the model. R.K.F. and A.A. wrote the paper with help from A.M.

## Funding

## Competing interests

The authors declare no competing interests.
