## [Peer Review File · Nature Communications]

REVIEWER COMMENTS

Reviewer #1 (Remarks to the Author):

The mitochondrial ATP synthase generates ATP in eukaryotes, and rows of dimeric ATP synthase induce curvature in membrane which is important for the function of mitochondria. While other types of ATP synthase have been well characterized structurally, the structural basis of type III ATP synthase, which is present in ciliates, is not well understood. In this manuscript, the authors present single-particle cryo-EM analysis of the *T. thermophila* ATP synthase purified in digitonin. The cryo-EM structures of type III ATP synthase in its dimeric and tetrameric forms revealed the basis of oligomerization of type III ATP synthase, and identified its unique protein components, co-factors and specific lipid molecules mediating protein interactions. The architecture of tetrameric ATP synthase indicate how it induces the membrane curvature. The cryo-EM work was nicely done, and the manuscript well written with excellent figures. Their findings add rich detail and provide new insights of ATP synthase, and would be interesting to the researchers in the related areas. I have a few suggestions and questions listed below.

(1) Does *T. thermophila* have only type III ATP synthase? Did other types of ATP synthase appear in single-particle analysis, or previous cryo-ET study?

(2) The authors nicely describe the symmetry mismatch in dimeric ATP synthase due to the presence of ATPPT2 and other components. However, it is unclear how much the main bodies of the two ATP synthases (e.g. F1 and central stalk) deviate from a perfect C2 symmetrical architecture. Can the authors clarify this?

Is there any major difference in the structure between the two enzymes within the dimer? Are they in the same rotatory state?

(3) In the tetramer structure, the angle (11 degree) and distance (140 Å) between the two dimers are important for concluding how membrane curvature can be generated. However, the actual angle and distance likely vary in a certain range. To gain insights of relative movement between the two dimers, multibody refinement or similar heterogeneity analyses should be performed.

(4) It is unclear how the arrangement of ATP synthase in Fig. 7 was generated. Is this a pure theoretical model based on the angle/distance between two dimers in the final tetramer cryo-EM map? Is this based on previously published in situ low-resolution structure of tubular membrane?

Other minor points:

Paragraph starting with line 167: The comparison of IF1 observed in ciliate ATP synthase with mammalian IF1 is particularly interesting. Some discussion in lines 416-432 can be moved here.

Fig. 1f, Fig. 4b: It is not obvious which portion and view of these panels related to the entire structure. Some extra labels would be helpful to orient readers.

Fig. 5: It would be helpful to make this figure larger to clearly see the lipids and labels.

In Fig. 1a and Fig. 6a, "local-resolution filtered cryo-EM density map". The program/method to filter cryo-EM maps according to local resolution should be clearly described in the method section.

ED Fig. 1b: 2D class averages have a strange coloring scheme. This seems confusing and provides no extra information compared to traditional white-on-black style.

ED Fig. 1c: The green class (class V) is tetramer. Why and how can it be processed as dimer?

Reviewer #2 (Remarks to the Author):

Flygaard et al has used the technique of single particle cryoEM to determine the structure of the mitochondrial ATP synthase dimer/tetramer from the single celled Eukaryote *Tetrahymena thermophila*. The structure is impressive and the data technically sound. However, the manuscript as written is extremely dense and inaccessible to the general readership of Nature communications. Also there is a lack of clarity about the significance of the structure from a biological and functional view point. It is a great structure, but what new biological information does it provide which will advance our knowledge of ATP synthase function and make the structure interesting to the reader and a challenge current thinking in the field?

The principle problem with the manuscript is the authors focus on minute details of how different subunits interact before giving a general overall description of the complex. Thus even as a ATP synthase specialist I found it difficult to follow what the authors had written, and the significance of the details they describe. Basically to really understand the structure, I need to download the PDB and look and interpret the structure myself as I failed to obtain a descent comprehension of the structure from the manuscript. Thus for the manuscript to be published in Nature communication, the text needs to be simplified or made clearer so people unfamiliar (and those that are familiar) can follow the authors descriptions and understand what is new and the significance of those findings.

My principle recommendation are:

- 1) Please describe what is identical in the new structure to previously solved ATP synthase dimers. An RMSD analysis and a supplementary figure highlighting the similarities and differences of the conserved subunits would be greatly appreciated. This should occur near the start of the results section so the readers can orient themselves with the structure based on what they previously know.
- 2) Once the conserved subunits are explained, please give an global overview of the additional subunits, where they occur e.g. induce the concept of wings, matrix region, luminal region etc and which subunits are found in these different regions. Once this is done, a more detailed description of each region can be provided.
- 3) In regards to the new subunits, I would appreciate a description of how the names were assigned. Also a table indicating the closest homologous protein and location in the structure would also aid comprehension.
- 4) once the global overview is described, each section described in 2 can be explained in detail. However, rather than just describing which subunit interacts with which subunit, the descriptions should be based more on proposed function and significance of interaction e.g. why is all this detail important, what is the significant of interaction a to interaction b. Just a simple description of what interacts with what is pretty boring and does not provide information about the biological role or significance of the structure and how it works.

In conclusion, the structure is impressive, but it is unclear why this structure would be of interest to the Nat Comm readership. Furthermore, to make the manuscript accessible and the important biological significant findings obvious, the authors need to describe the structure in general terms and compare to existing structures before focusing down into minute details of which subunit interacts with which.

Reviewer #3 (Remarks to the Author):

Flygaard and coworkers present near atomic-resolution cryo-EM structures of type III ATP synthase from the ciliated protozoan, *Tetrahymena thermophila*. Unlike ATP synthases from mammalian or algae mitochondria, which form V-shaped dimers, type III enzymes are U-shaped. The characteristic U-shape is proposed to stabilize the tubular cristae membranes of *T. thermophila* mitochondria. Prior proteomics work indicated the presence of multiple subunits unique to the *T. thermophila* enzyme, the architecture and function of which are not known. The authors obtained maps of dimeric and tetrameric ATP synthases at overall resolutions between 2.5 - 3.1 Å (for the dimer), and 3.3 Å for the tetramer. These maps allowed the authors to model 28 different subunits, including nine novel polypeptides that hadn't been identified before. Interestingly, the C2 symmetry that is observed for dimeric ATP synthase from animal or algae mitochondria is broken by a single copy subunit (ATPTT2) that serves as an anchor for the C-termini of dimeric inhibitor protein. Besides the canonical ATP synthase subunits, the authors identified and modeled several novel associated proteins, such as ATPTT1, a subunit homologous to mitochondrial sulfide:quinone oxidoreductase. They were also able to model various lipid and tightly bound coenzyme Q molecules as well as a molecule of MgATP bound to the Fo.

Overall, the study does provide a significant amount of novel high-quality information on the structure of the unique membrane subunits of the ciliate enzyme, how these membrane subunits interact to form dimers and tetramers, and how the tetramers oligomerize to shape the tubular cristae. The authors also uncover a number of associated proteins and cofactors not seen previously associated with ATP synthase. Thus, from a technical standpoint, this "tour-de-force" appears to be carried out with great care and will certainly be of interest to the rotary ATPase field. However, enthusiasm is somewhat dampened by the highly descriptive nature of the results section. What seems to be somewhat missing is a bridge between the structural information, and any potentially exciting and novel insight into the mechanism and specific function of this type III ATP synthase. While the base mechanism of ATP synthesis is likely conserved between the different type ATP synthases, interesting mechanistic questions could include: why does the type III enzyme have all the extra membrane subunits?, what is the function of these extra subunits?, and what is the advantage for this organism to use these U-shaped structures over the conventional V-shaped dimers found in other organisms? Moreover, a closer inspection of the modeling reveals that the "2.7 Å resolution cryo-EM structure" of the dimer appears to be an assembly of individual domains resolved at various resolutions using focused classification. While this approach appears to be now common practice in the molecular motor cryo-EM field, calling the mosaic a "2.7 Å structure of the dimer" seems to be misleading as it gives the impression that near atomic information is obtained for one structural state of the motor as a whole. Looking at Extended data fig. 2, it is clear that the "2.7 Å" model of the dimer is really sub-3 Å only for the membrane bound 'stator' of the motor, with the c-ring rotor and F1 part resolved at only >4-6 Å. Even the model of the supposedly 2.5 Å membrane domain does not contain the rotor (because it does not adopt a defined rotational position with respect to the 'a' subunit?). A likely explanation for the low resolution of the rotor is that mitochondrial ATP synthase can exist in multiple rotary states, however, this is not even mentioned in the manuscript.

Major points:

- (1) Do the preparations (dimer and tetramer) have ATPase activity? If so, is the activity sensitive to e.g. oligomycin or DCCD?
- (2) Please show SDS-PAGE of the dimer and tetramer preparations.
- (3) It is acknowledged that describing and interpreting a multi-subunit complex the size of Tt-ATP synthase dimer is a daunting task, but as it stands, much of the results and figures is descriptive in nature, and therefore difficult to follow due to huge number of polypeptides that constitute the dimer. Sometimes less is more, and instead of trying to show every polypeptide individually and from different angles (e.g. Figure 2), which in itself carries very little interesting insight, it may be better to

focus on structural features that can be related to special aspects of the function and mechanism of this complex.

(4) Figure 1: Panels a, b are OK. Panels c, e, f - difficult to extract interesting information. Especially f - we are supposed to see a "cavity extending towards c-ring" - where is the c-ring? Some annotation would be helpful here (this is also true for many other figure panels).

(5) Related to point (2), page 3, lines 114-6: "This Fo-channel likely provides access for bulk membrane lipids to the c-ring, sealing off access from the luminal side thereby establishing conditions for proton translocation" If access from lumen to c-ring is "sealed off", how are protons (H³O⁺) gaining access to the essential glutamates of the c-ring?

(6) Extended data fig. 2: Map resolution of dimer is given as 2.7 Å. However, the reconstruction is dominated by the membrane stator portion, with little contribution of the F1 and central rotor to the FSC. Is it fair to call this a "2.7 Å resolution structure of the dimer"?

(7) Related to (4), there is no mentioning of rotary states of the motor. This is very different in comparison to the enzyme from e.g. *Polytomella* (ref. 21). Did the authors not find different rotary states? Is there anything they can say about nucleotide content of F1?

(8) Page 7 - Can the authors say a little bit more about the function of the sulfide:quinone oxidoreductase and why it may be part of ATP synthase?

Minor points:

(9) Figure 3: Residue labels are difficult to make out. Also applies to figures 4 and 5.

(10) page 5, lines 164-166: "...binding of ATP_{2TT2} induces different conformational changes..." Maybe better to say "...induces different conformations..."?

(11) Page 9, lines 253/254 "...largest subunit-a in any ATP synthase..." The authors may want to qualify this statement as 'a' subunits in A- and A/V-like ATP synthases are significantly larger.

(12) Page 9, lines 273-278: "Type III ATP synthase dimer contains the most extensive dimer interface..." How do the "16,000 Å² compare to e.g. the *polytomella* dimer?

(11) Page 12, lines 343-345: "...tetramer complexes of the photosynthetic membrane"? Please clarify.

We thank the reviewers for kindly providing a thorough analysis of our data and constructive comments that helped to improve the manuscript. The reviewers brought up a number of important points primarily regarding the data presentation and the layout of the paper. Each one of the points has been addressed as specified below. Particularly, the manuscript has been restructured and shortened, three new figures and a movie have been added to better illustrate the differences of the overall structure from other ATP synthases and the unique features of IF₁. All figures have been revised according to the guideline provided by the reviewers. Multiple formulations improved throughout the text to emphasize the functional aspects. Following the same considerations, the title has also been changed. Native BN-PAGE, multi-body refinement analysis and multiple-sequence alignment have additionally been performed. Below is the detailed point-by-point response.

Reviewer #1 (Remarks to the Author):

The mitochondrial ATP synthase generates ATP in eukaryotes, and rows of dimeric ATP synthase induce curvature in membrane which is important for the function of mitochondria. While other types of ATP synthase have been well characterized structurally, the structural basis of type III ATP synthase, which is present in ciliates, is not well understood. In this manuscript, the authors present single-particle cryo-EM analysis of the *T. thermophila* ATP synthase purified in digitonin. The cryo-EM structures of type III ATP synthase in its dimeric and tetrameric forms revealed the basis of oligomerization of type III ATP synthase, and identified its unique protein components, co-factors and specific lipid molecules mediating protein interactions. The architecture of tetrameric ATP synthase indicate how it induces the membrane curvature. The cryo-EM work was nicely done, and the manuscript well written with excellent figures. Their findings add rich detail and provide new insights of ATP synthase, and would be interesting to the researchers in the related areas. I have a few suggestions and questions listed below.

(1) Does *T. thermophila* have only type III ATP synthase? Did other types of ATP synthase appear in single-particle analysis, or previous cryo-ET study?

- As requested, we now added this information on page 3, lines 94-94, and referred to previous studies: "no other types were detected that is consistent with previous low resolution studies^{14,23}"

(2) The authors nicely describe the symmetry mismatch in dimeric ATP synthase due to the presence of ATPPT2 and other components. However, it is unclear how much the main bodies of the two ATP synthases (e.g. F1 and central stalk) deviate from a perfect C2 symmetrical architecture. Can the authors clarify this?

Is there any major difference in the structure between the two enzymes within the dimer? Are they in the same rotatory state?

- As requested, we now added a clarification on page 6, lines 189-190: “The association of the ATPPT2 subunit enforces local deviations from the symmetry of the dimer, while the main part of the F_o -subcomplex obeys the symmetrical architecture”; and on page 3, lines 91-92: “Both monomers are inhibited in the same rotary state by the natural inhibitor IF1.”

Because of the intra-dimeric IF1 linking of the F1-subcomplexes, they would indeed be related by C2 symmetry. However, as we looked to see whether we had other rotational states by symmetry-expanding particles, subtracting one half, aligning on F_o and classifying F1/c-ring without alignment, while we did not see any other rotational states, we noticed there is limited flexibility in the c-ring position with respect to F_o and central stalk. This intrinsic flexibility accounts for the deteriorated map quality of those regions in the consensus dimer map.

In addition, we have added a supplementary movie 1 which shows that the dimer has one single rotational state.

(3) In the tetramer structure, the angle (11 degree) and distance (140 Å) between the two dimers are important for concluding how membrane curvature can be generated. However, the actual angle and distance likely vary in a certain range. To gain insights of relative movement between the two dimers, multibody refinement or similar heterogeneity analyses should be performed.

- As suggested by the reviewer, we performed a multi-body refinement analysis on the ATP synthase tetramer particles, masking the two dimers separately and treating them as bodies. The result was a rather complex motion landscape, where unfortunately no single component was found to describe the majority of the continuous, intrinsic motion. We have included a Supplementary Fig. 14 to show the results of the multi-body analysis and the motions described by the first three eigenvectors. We have also added a section in the main text describing the analysis and our results on page 13, lines 300-306.

(4) It is unclear how the arrangement of ATP synthase in Fig. 7 was generated. Is this a pure theoretical model based on the angle/distance between two dimers in the final tetramer cryo-EM map? Is this based on previously published in situ low-resolution structure of tubular membrane?

- The helix was generated by arranging ATP synthase tetramer density maps using a sliding tetramer map for aligning neighbouring ATP synthases. The geometry is entirely based on the intrinsic geometry of the tetramer reconstruction. We added a clarification in the figure legend that the proposed arrangement represents an idealised model, and is not based on the previously published structure: “Helical row model of type III ATP synthase extrapolated from the single-particle reconstruction”.

Other minor points:

Paragraph starting with line 167: The comparison of IF1 observed in ciliate ATP synthase with mammalian IF1 is particularly interesting. Some discussion in lines 416-432 can be moved here.

- *As suggested, we added a comparative analysis in the text, as well as two new panels in Figure 3, comparing the IF1 and its interactions with the mammalian counterpart.*

Fig. 1f, Fig. 4b: It is not obvious which portion and view of these panels related to the entire structure. Some extra labels would be helpful to orient readers.

- *We have added a black outlined window in Fig. 1b to highlight the region for the close-up view in Fig. 1f. Similarly, we have indicated in Fig. 4a with a black window the region for the close-up view in Fig. 4b. Figure legends have been updated to mention these highlighting windows for close-up views. We also improved the presentation in most of the other figures according to the reviewer's guideline.*

Fig. 5: It would be helpful to make this figure larger to clearly see the lipids and labels.

- *We have updated the figure increasing the figure size and labels. Also, we have added labels on bound cardiolipins. We also increased the label size in other figures.*

In Fig. 1a and Fig. 6a, “local-resolution filtered cryo-EM density map”. The program/method to filter cryo-EM maps according to local resolution should be clearly described in the method section.

- *We have added a sentence in the Methods section describing that local-resolution filtered maps were generated using RELION-3.1.*

ED Fig. 1b: 2D class averages have a strange coloring scheme. This seems confusing and provides no extra information compared to traditional white-on-black style.

- *We have changed the coloring scheme of 2D class averages to white-on-black.*

ED Fig. 1c: The green class (class V) is tetramer. Why and how can it be processed as dimer?

- *We regret the confusion caused in the figure of the 3D classes. The original 3D class V with tetramer particles was not included for dimer reconstruction, only particles from original classes IV and VI. We have now changed the figure so that 3D classes with tetramer particles are kept to the left as classes I-IV and dimer particles to the right as classes V and VI.*

Reviewer #2 (Remarks to the Author):

Flygaard et al has used the technique of single particle cryoEM to determine the structure of the mitochondrial ATP synthase dimer/tetramer from the single celled Eukaryote *Tetrahymena thermophila*. The structure is impressive and the data technically sound. However, the manuscript as written is extremely dense and inaccessible to the general readership of Nature communications. Also there is a lack of clarity about the significance of the structure from a biological and functional view point. It is a great structure, but what new biological information does it provide which will advance our knowledge of ATP synthase function and make the structure interesting to the reader and a challenge current thinking in the field?

The principle problem with the manuscript is the authors focus on minute details of how different subunits interact before giving a general overall description of the complex. Thus even as a ATP synthase specialist I found it difficult to follow what the authors had written, and the significance of the details they describe. Basically to really understand the structure, I need to download the PDB and look and interpret the structure myself as I failed to obtain a descent comprehension of the structure from the manuscript. Thus for the manuscript to be published in Nature communication, the text needs to be simplified or made clearer so people unfamiliar (and those that are familiar) can follow the authors descriptions and understand what is new and the significance of those findings.

- We now simplified the text, restructured the manuscript, removed some of the minute details from the text, and moved corresponding figure panels to Supplementary. The revised manuscript has been shortened by more than 20%, and new findings have been emphasized based on the reviewer's comments. A paragraph at the end of the Introduction section has been revised to provide an overview of the structural data, and additional changes are specified below.

My principle recommendation are:

1) Please describe what is identical in the new structure to previously solved ATP synthase dimers. An RMSD analysis and a supplementary figure highlighting the similarities and differences of the conserved subunits would be greatly appreciated. This should occur near the start of the results section so the readers can orient themselves with the structure based on what they previously know.

- We introduced a paragraph on page 3 in the start of the results section, exactly as suggested by the reviewer, and added the requested illustration as Supplementary Figure 5. We also added an overview of the interactions between the monomers as Supplementary Figure 6. RMSD analysis unfortunately turned out not to be informative, because of the large conformational changes, induced by the new subunits.

2) Once the conserved subunits are explained, please give an global overview of the additional subunits, where they occur e.g. induce the concept of wings, matrix region, luminal region

etc and which subunits are found in these different regions. Once this is done, a more detailed description of each region can be provided.

- Following the suggestion, the description has been added after the global overview on pages 3-4. Supplementary movie 1 is added to further clarify the details of the different regions and show it in the context of the overall structure. Supplementary Figure 5, 6 and 7 illustrates how the additional subunits affect the overall organisation and features the differences from the previously reported structural data.

3) In regards to the new subunits, I would appreciate a description of how the names were assigned. Also a table indicating the closest homologous protein and location in the structure would also aid comprehension.

- The information about the nomenclature has been added on page 3, lines 104-106. The location in the structure is indicated in Figure 1b. The individual structures of each protein are shown in the new Supplementary Figure 5. The densities for each individual newly identified protein are shown in Supplementary Figure 4. The information regarding the location in the structure, number of residues (built and total) for each protein, used gene name and protein ID is given in Supplementary Table 2.

We also performed the requested phylogenetic analysis, however it shows that most of the newly identified proteins in the type III ATP synthase are specific to the subset of organisms with type III ATP synthase. We provide the table below as it is unfortunately not informative enough to be included in the manuscript.

	subunit-a	subunit-b	subunit-d	subunit-f	subunit-8	subunit-l/j	subunit-k	ATPT1	ATPT2	ATPT3	ATPT4	ATPT5	ATPT6	ATPT7	ATPT8	ATPT9	ATPT10	ATPT11	ATPT12	ATPT13	
Ciliates																					
Tetrahymena thermophila	x	x	x	x	x	x	x	x	x	x	x	x	x	x	x	x	x	x	x	x	x
Paramecium tetraurelia	x	x	x	x	x	x	x	x	x	x	x	x	x	x	x	x	x	x	x	x	x
Ichthyophthirius multifiliis	x	x	x	x	x	x	x	x	x	x	x	x	x	x	x	x	x	x	x	x	x
Pseudocohnilembus persalinus	x	x	x	x	x	x	x	x	x	x	x	x	x	x	x	x	x	x	x	x	x
Stentor coeruleus	x	x	(-)	x	x	x	(-)	x	x	(-)	(x)	(x)	(-)	x	x	x	x	x	(-)	x	x
Stylonychia lemnae	x	x	x	(-)	(-)	x	x	x	x	x	x	x	x	x	x	x	x	x	x	x	x
Halteria grandinella	x	x	(-)	(-)	(x)	(x)	x	(x)	x	(x)	(-)	x	x	x	(-)	(x)	(x)	(x)	(-)	x	x

Green: presence of homologous protein; yellow: there might be a candidate homologous protein;

red: no homologous protein found

4) once the global overview is described, each section described in 2 can be explained in detail.

However, rather than just describing which subunit interacts with which subunit, the descriptions should be based more on proposed function and significance of interaction e.g. why is all this detail important, what is the significant of interaction a to interaction b. Just a simple description of what interacts with what is pretty boring and does not provide information about the biological role or significance of the structure and how it works.

- We restructured the manuscript as suggested and removed overly descriptive parts. The revised version of the now follows the suggested order and content. Following the overall description of the parallel dimer, the ms describes the characteristics of the specific subunits (page 5). Then it introduces ATPTT2 as a symmetry-breaking element emphasising a putative function of anchoring IF₁ (page 6). This represents the first report of sub-stoichiometric

subunit in any ATP synthase dimer. It is also the first observation of IF₁ anchoring. Next, the text features subunit-a as a dimer-forming element (pages 8-9). Finally, we explain that dimers are not sufficient for membrane curvature and introduce the data on the tetramer that is also linked to subunit-a (page 11). The ms concludes with the discussion that leads to the rationalisation of the tetramer as the functional unit that propagates membrane curvature. In addition, the title has been changed to highlight the biological role.

In conclusion, the structure is impressive, but it is unclear why this structure would be of interest to the Nat Comm readership. Furthermore, to make the manuscript accessible and the important biological significant findings obvious, the authors need to describe the structure in general terms and compare to existing structures before focusing down into minute details of which subunit interacts with which.

- We hope the Reviewer finds the revised version more appropriate, as we tried to follow his/her guideline point by point. Only the structural details leading to conclusions are mentioned in the manuscript. Since this is a research article, a more detailed comparative analysis would probably be outside the scope of the journal, however we now added comparisons of some function-related aspects to the existing structures in the new figures, including the main Figure 3.

Reviewer #3 (Remarks to the Author):

Flygaard and coworkers present near atomic-resolution cryo-EM structures of type III ATP synthase from the ciliated protozoan, *Tetrahymena thermophila*. Unlike ATP synthases from mammalian or algae mitochondria, which form V-shaped dimers, type III enzymes are U-shaped. The characteristic U-shape is proposed to stabilize the tubular cristae membranes of *T. thermophila* mitochondria. Prior proteomics work indicated the presence of multiple subunits unique to the *T. thermophila* enzyme, the architecture and function of which are not known. The authors obtained maps of dimeric and tetrameric ATP synthases at overall resolutions between 2.5 - 3.1 Å (for the dimer), and 3.3 Å for the tetramer. These maps allowed the authors to model 28 different subunits, including nine novel polypeptides that hadn't been identified before. Interestingly, the C2 symmetry that is observed for dimeric ATP synthase from animal or algae mitochondria is broken by a single copy subunit (ATPTT2) that serves as an anchor for the C-termini of dimeric inhibitor protein. Besides the canonical ATP synthase subunits, the authors identified and modeled several novel associated proteins, such as ATPTT1, a subunit homologous to mitochondrial sulfide:quinone oxidoreductase. They were also able to model various lipid and tightly bound coenzyme Q molecules as well as a molecule of MgATP bound to the F_o.

Overall, the study does provide a significant amount of novel high-quality information on the structure of the unique membrane subunits of the ciliate enzyme, how these membrane subunits interact to form dimers and tetramers, and how the tetramers oligomerize to shape the tubular cristae. The authors also uncover a number of associated proteins and cofactors

not seen previously associated with ATP synthase. Thus, from a technical standpoint, this “tour-de-force” appears to be carried out with great care and will certainly be of interest to the rotary ATPase field.

However, enthusiasm is somewhat dampened by the highly descriptive nature of the results section. What seems to be somewhat missing is a bridge between the structural information, and any potentially exciting and novel insight into the mechanism and specific function of this type III ATP synthase. While the base mechanism of ATP synthesis is likely conserved between the different type ATP synthases, interesting mechanistic questions could include: why does the type III enzyme have all the extra membrane subunits?

what is the function of these extra subunits?

and what is the advantage for this organism to use these U-shaped structures over the conventional V-shaped dimers found in other organisms?

- We have rewritten the text to highlight that the type III subunits are responsible for holding the two monomers in parallel to each other (page 3, last paragraph; dedicated section on pages 8-9). The revised version further explains that the U-shaped structure does not induce membrane curvature, which is therefore achieved by the tetramer (dedicated section on page 12).

Moreover, a closer inspection of the modeling reveals that the “2.7 Å resolution cryo-EM structure” of the dimer appears to be an assembly of individual domains resolved at various resolutions using focused classification. While this approach appears to be now common practice in the molecular motor cryo-EM field, calling the mosaic a “2.7 Å structure of the dimer” seems to be misleading as it gives the impression that near atomic information is obtained for one structural state of the motor as a whole. Looking at Extended data fig. 2, it is clear that the “2.7 Å” model of the dimer is really sub-3 Å only for the membrane bound ‘stator’ of the motor, with the c-ring rotor and F1 part resolved at only >4-6 Å. Even the model of the supposedly 2.5 Å membrane domain does not contain the rotor (because it does not adopt a defined rotational position with respect to the ‘a’ subunit?). A likely explanation for the low resolution of the rotor is that mitochondrial ATP synthase can exist in multiple rotary states, however, this is not even mentioned in the manuscript.

- We apologize for the confusion and now deleted the 2.7 Å resolution value from the abstract. 2.7 Å is the overall resolution of the dimer after 3D refinement and classification, and it is shown in the Supplementary Figure 2 (right panel). The other parts were resolved upon masked refinement as follows: F_o - 2.5 Å, wing – 2.8 Å, rotor - 2.9 Å, stalk/c-ring – 3.1 Å. The information is provided in the Supplementary Figure 2 (right panel, bottom). We now explain in detail in the first paragraph of the Results: “Using masked refinements, the resolution of different regions reached 2.5-3.1 Å (Supplementary Fig. 2,3a-e and Supplementary Table 1), allowing de novo modelling. Refinement into a 2.7-Å resolution consensus map allowed composite model construction of the ATP synthase dimer. The tetramer was refined to 3.3 Å resolution and the membrane region consisting of interacting F_o subcomplexes was further improved to 3.1 Å resolution upon masked refinement

(Supplementary Fig. 2,3f). Two copies of the ATP synthase dimer model were refined into the F_o-tetramer map generating a composite ATP synthase tetramer model.”

Only one state has been detected for the rotor due to the presence of the IF₁ inhibitor. We now clarified this information in the related Supplementary Figures 2 and 3, and the text, where it is also stated that part of IF₁ could not be modeled due to its flexibility (lines 207-208). Furthermore, for the newly discovered subunits, we show the density map in Supplementary Figure 4, so that a reader can judge the quality of the model building and reliability of the identification.

Major points:

(1) Do the preparations (dimer and tetramer) have ATPase activity? If so, is the activity sensitive to e.g. oligomycin or DCCD?

- The ATP synthase has been isolated in its inhibited form. Therefore, no enzymatic activity has been detected.

(2) Please show SDS-PAGE of the dimer and tetramer preparations.

- Since the preparation includes contaminating mitochondrial complexes, such as pyruvate dehydrogenase, α -ketoglutarate dehydrogenase, and others, the SDS-PAGE has not been informative. Therefore, to comply with the Reviewer's request to provide an additional analysis, we assessed the preparation with blue-native PAGE, which is now presented in the Supplementary Figure 1. The BN-PAGE analysis was initially performed to assess solubilization efficiency and preservation of the high-order complexes to the ATP synthase dimer band. A description of this has been added on page 3, and the experiments described in Methods on page 16.

(3) It is acknowledged that describing and interpreting a multi-subunit complex the size of Tt-ATP synthase dimer is a daunting task, but as it stands, much of the results and figures is descriptive in nature, and therefore difficult to follow due to huge number of polypeptides that constitute the dimer. Sometimes less is more, and instead of trying to show every polypeptide individually and from different angles (e.g. Figure 2), which in itself carries very little interesting insight, it may be better to focus on structural features that can be related to special aspects of the function and mechanism of this complex.

- We regret that the first version of the manuscript and illustrations read too descriptive. The revised version has been generally improved based on the comments. Specifically, regarding Figure 2, we improved all the panels, and focused attention on the binding site of ATPTT2, and differences between the binding subunits from the two monomers. In the revised version surface representation, and transparency are employed to tone down the less interesting insight. Also, we have added a supplementary movie 1 to provide better representation of the subunits in Figure 2.

(4) Figure 1: Panels a, b are OK. Panels c, e, f - difficult to extract interesting information.

Especially f - we are supposed to see a “cavity extending towards c-ring” - where is the c-ring? Some annotation would be helpful here (this is also true for many other figure panels).

- *We have added a label showing the location of the c-ring with respect to the F_o cavity. (See also the response to Reviewer #1 (minor point 2). Added connection between the panels. In addition, we have updated all figures with increased label font sizes and added annotations for cardiolipins in Fig. 5b,c.*

(5) Related to point (2), page 3, lines 114-6: “This F_o -channel likely provides access for bulk membrane lipids to the c-ring, sealing off access from the luminal side thereby establishing conditions for proton translocation” If access from lumen to c-ring is “sealed off”, how are protons (H_3O^+) gaining access to the essential glutamates of the c-ring?

- *The lipid access to the c-ring only establishes the matrix proton half-channel, which must be sealed off from the crista lumen so as to not short-circuit the system. We have rephrased the sentence: “This F_o cavity likely provides access for bulk membrane lipids to the c-ring, establishing the lumenally sealed matrix proton half-channel required for translocation of protons” (page 4, lines 126-128).*

(6) Extended data fig. 2: Map resolution of dimer is given as 2.7 Å. However, the reconstruction is dominated by the membrane stator portion, with little contribution of the F_1 and central rotor to the FSC. Is it fair to call this a “2.7 Å resolution structure of the dimer”?

- *Although all the technical details were described according to the standards in the cryo-EM field, and F_1 subunits are not discussed in the paper, we now removed the “2.7 Å resolution” from the abstract. To be open with a reader, the first paragraph of the Results section explains how the resolution was attained for the different parts: “We used masked refinements of separate regions of the ATP synthase dimer, which resulted in 2.5-3.1 Å-resolution maps (Supplementary Fig. 2,3a-e and Supplementary Table 1), allowing de novo modelling of the respective regions. Refinement into a 2.7-Å resolution consensus map allowed composite model construction of the ATP synthase dimer.”*

In addition, we also added to the figure titles for each panel showing the masked reconstructions. Please note that the figure illustrates only the surface area of the masked elements, whereas most of the described findings are at the membrane core, where the local resolution is higher than 2.7 Å.

(7) Related to (4), there is no mentioning of rotary states of the motor. This is very different in comparison to the enzyme from e.g. *Polytomella* (ref. 21). Did the authors not find different rotary states? Is there anything they can say about nucleotide content of F_1 ?

- *Since our structure contains IF_1 , no rotary states are detected. In *Polytomella*, there is no IF_1 , which was the reason the authors chose that organism to investigate rotary states. To avoid confusion for a reader, we now added the information upfront, in the first paragraph describing the results, on page 3, lines 91-92: “Both monomers are inhibited in the same rotary state by the natural inhibitor IF_1 .” The nucleotide content of F_1 is as expected: ATP bound in all three alpha subunits, and ADP bound betaDP and betaTP subunits, no*

nucleotide bound in betaE. Since the data provides no new information, we decided not to include its description in the manuscript.

(8) Page 7 - Can the authors say a little bit more about the function of the sulfide:quinone oxidoreductase and why it may be part of ATP synthase?

- Please see the Results section on page 5, lines 148-158; and Discussion section on page 15, lines 354-359.

Minor points:

(9) Figure 3: Residue labels are difficult to make out. Also applies to figures 4 and 5.

- We have updated all figures with increased label font sizes, and improved the layout for Figure. 5.

(10) page 5, lines 164-166: "...binding of ATP2TT2 induces different conformational changes..." Maybe better to say "...induces different conformations..."?

- Changed as suggested.

(11) Page 9, lines 253/254 "...largest subunit-a in any ATP synthase..." The authors may want to qualify this statement as 'a' subunits in A- and A/V-like ATP synthases are significantly larger.

- Added: "F-type".

(12) Page 9, lines 273-278: "Type III ATP synthase dimer contains the most extensive dimer interface..." How do the "16,000 Å² compare to e.g. the polytomella dimer?"

- As we have not found a deposited, complete dimer model of the Polytomella ATP synthase, we used PDB 6RD4 model and fitted two copies into EMD-4805 and merged the two copies into a single model file for dimer interface analysis using PISA server. The result was ~4000 Å² of buried surface area in the Polytomella ATP synthase dimer. However, since we generated the composite dimer model of Polytomella ATP synthase only by fitting models into the composite EM density map, and not performing model refinement against an experimental map, deviations from the calculated buried surface area can be expected. To comply with the Reviewer's request, we added to the text: "which is at least 4 times more ..."

(11) Page 12, lines 343-345: "...tetramer complexes of the photosynthetic membrane"? Please clarify.

- We have deleted this comparison to avoid confusion.

REVIEWER COMMENTS

Reviewer #1 (Remarks to the Author):

All my comments have been adequately addressed. I support its publication in Nature Communications.

Reviewer #2 (Remarks to the Author):

Flygaard et al., have made an excellent effort at rewriting their manuscript entitled "Type III ATP synthase is a symmetry-deviated dimer that induces membrane curvature through tetramerization". The manuscript was much easier and more enjoyable to read. It was easier to follow the authors' description of the structure and the biological relevance. The new text has raised a few additional scientific questions and the figures, especially the main text, still need to be made more transparent so the reader can understand what they are meant to be looking at. Overall, the manuscript is greatly improved and would be of interest to the readership of nature communications.

Point-by-point comment:

Line 95: The authors state "The ~2 MDa complex comprises 81 protein subunits, 9 of which were not previously reported, and assigned directly from our cryo-EM density map". Which subunits were previously reported and where were they reported?

Page 5: ATPPT1 is described in detail but the biological role/purpose is not clear. Additional information is required. Has the authors attempted functional assays to assess if the subunit has sulfide:quinone oxidoreductase activity? If not, why not? If no activity is observed, what has changed in the subunit from the active enzyme? How tightly is the subunit bound to the complex and how essential is that subunit? Its current position doesn't seem to provide any structural support to the complex. Can it be deleted without affecting the function of the enzyme complex or membrane curvature? In a previous manuscript Muehleip has suggested the ATPPT1 density is important in initiating curvature of the lumen. Is this suggestion still valid based on this higher-resolution structure?

Dimer-dimer interface: How strong is this interface? What distinguishes the described interface from the interface that might be formed on the other side of the dimer? Is the lipid/protein interface in the membrane sufficient to hold two dimers together without IF1? If so, how was this tested? What is the benefit of having a protein-protein interaction in the membrane as opposed to two dimers coming together based on protein concentration and optimal packing? What strength of interaction would be required to hold two dimers together? E.g. How significant is the interaction between dimers that you describe? Is it really strong enough to have a functional role or is it more that they are just located next to each other?

In Fig3: The authors shown that two IF1 bind to two monomers where each monomer belongs to a different dimer in the same tetramer? What causes the IF1 to bind to the monomers in the same set of dimers? Can the IF1 join different dimers together? Is the IF1 the reason why you get tetramers in your purification? Would you still get tetramers if you delete the IF1?

Dimer interface: the main text figures and supplementary fig 6 do not show the dimer interface clearly or informatively to the reader. In Fig 5, panel a is a mismatch of colors. For the box labelled b "off-site", how is this contributing to the interface, which helices come from which dimer. What is the significance of panels B, c and d? what are we meant to be observing here in relation to the function of the enzyme? What is the take home message from the figure? Supplementary figure 6 would benefit from additional panels. E.g. one panel showing the monomer from the orientation of the

interface, with the subunits which participate in the interface colored and labeled.

Fig 4: It would be helpful if the C and N-terminal ends of subunit a are labelled in panel a. How does panel b relate to panel a? What is the difference between fig 4 panel D boxed region and Fig 1 panel f?

Fig 7: how do panels b and c relate to panel a and each other? Arrows indicating the degree of rotation would be helpful. For panel d, where is this taken from in relation to panel a? what exactly are we looking at and where is the protein-protein interaction located that is displayed in the other three panels?

I really like supplementary figure 5.

Reviewer #3 (Remarks to the Author):

In the revised manuscript, the authors have amended the text, improved figure layout and annotations, and they added one new figure. With these revisions, and the written response in their rebuttal, the authors were able to address all queries and critiques of the first round of review in a satisfactory manner. Overall, the revised manuscript is improved to a point where publication in Nature Communications can now be recommended.

Point by point response:

Line 95: The authors state "The ~2 MDa complex comprises 81 protein subunits, 9 of which were not previously reported, and assigned directly from our cryo-EM density map". Which subunits were previously reported and where were they reported?

It is stated in the text, in the corresponding table, and the reference is given on lines 59-60: "*Proteomics analysis detected some specific subunits with no sequence similarity to the other ATP synthase types (23)*"

Page 5: ATPPTT1 is described in detail but the biological role/purpose is not clear. Additional information is required. Has the authors attempted functional assays to assess if the subunit has sulfide:quinone oxidoreductase activity? If not, why not? If no activity is observed, what has changed in the subunit from the active enzyme? How tightly is the subunit bound to the complex and how essential is that subunit? Its current position doesn't seem to provide any structural support to the complex. Can it be deleted without affecting the function of the enzyme complex or membrane curvature? In a previous manuscript Muehleip has suggested the ATPPTT1 density is important in initiating curvature of the lumen. Is this suggestion still valid based on this higher-resolution structure?

The information is given on lines 152-158: "*The nicotinamide group of the NAD is in close proximity to Cys205 (Supplementary Fig. 9e), which could potentially be the active site of ATPPTT1. The human sulfide:quinone oxidoreductase is an integral membrane protein, and ATPPTT1 locates closely to the matrix leaflet of the crista membrane with several adjacent lipid-like density features resolved in the cryo-EM density map (Supplementary Fig. 9f). Thus, ATPPTT1 is positioned to encounter membrane-embedded electron-acceptors, such as CoQ, in a similar fashion as the human enzyme (26).*"

Figures 1A & B show that the detergent belt is bent in the periphery to accommodate ATPPTT1, which is located at half-height of the c-ring.

Dimer-dimer interface: How strong is this interface? What distinguishes the described interface from the interface that might be formed on the other side of the dimer? Is the lipid/protein interface in the membrane sufficient to hold two dimers together without IF1? If so, how was this tested? What is the benefit of having a protein-protein interaction in the membrane as opposed to two dimers coming together based on protein concentration and optimal packing? What strength of interaction would be required to hold two dimers together? E.g. How significant is the interaction between dimers that you describe? Is it really strong enough to have a functional role or is it more that they are just located next to each other?

The dimer interface has been described in detail on pages 6-11 with four main figures. Specifically, as stated the structure has C2 symmetry, and IF1 does not hold two dimers together. The questions on the assembly of the complex and evolutionary origin would be of interest, however this is outside the scope of the current research article featuring the unique characteristics of the type III ATP synthase.

In Fig3: The authors shown that two IF1 bind to two monomers where each monomer belongs to a different dimer in the same tetramer?

No, and it is specifically stated in the text that it is not the case, instead the IF1 binding is intra-dimeric, as shown and labelled in Figure 3.

What causes the IF1 to bind to the monomers in the same set of dimers?

The entire figure 3 is dedicated to this subject, and also provides a comparative analysis.

Can the IF1 join different dimers together? Is the IF1 the reason why you get tetramers in your purification? Would you still get tetramers if you delete the IF1?

We have not shown any data that would suggest such speculations.

Dimer interface: the main text figures and supplementary fig 6 do not show the dimer interface clearly or informatively to the reader. In Fig 5, panel a is a mismatch of colors.

It does not read to us as a constructive comment. All the figures have been revised based on the suggestions from the Reviewers. We do not detect a "mismatch of colors".

For the box labelled b "off-site", how is this contributing to the interface, which helices come from which dimer. What is the significance of panels B, c and d? what are we meant to be observing here in relation to the function of the enzyme? What is the take home message from the figure?

There is no such box in the figure, the correct quote is "off-axis", which is explained in the text. The significance of the figure is explained in the legend.

Supplementary figure 6 would benefit from additional panels. E.g. one panel showing the monomer from the orientation of the interface, with the subunits which participate in the interface colored and labeled.

We added the requested panels showing the interacting subunits and specific regions.

Fig 4: It would be helpful if the C and N-terminal ends of subunit a are labelled in panel a.

How does panel b relate to panel a? What is the difference between fig 4 panel D boxed region and Fig 1 panel f?

The rationale to avoid extra labelling was because the orientation of subunit-a is similar to the neighboring panel, where the termini are labelled. In addition, the figure already contains 8 labels. The relation between the panels is explained in the legend: "*Window corresponds to the region shown in panel b close-up view*".

Fig 7: how do panels b and c relate to panel a and each other? Arrows indicating the degree of rotation would be helpful. For panel d, where is this taken from in relation to panel a? what exactly are we looking at and where is the protein-protein interaction located that is displayed in the other three panels?

The same color code has been used throughout the four panels, so that it is easier for a reader to follow how the views are related.